# On the universality of deep learning

**Emmanuel Abbe**
Mathematics Institute
EPFL
Lausanne, 1005 Switzerland

**Colin Sandon**
Department of Mathematics
MIT
Cambridge, MA 02139

## Abstract

This paper shows that deep learning, i.e., neural networks trained by SGD, can learn in polytime any function class that can be learned in polytime by some algorithm, including parities. This universal result is further shown to be robust, i.e., it holds under possibly poly-noise on the gradients, which gives a separation between deep learning and statistical query algorithms, as the latter are not comparably universal due to cases like parities. This also shows that SGD-based deep learning does not suffer from the limitations of the perceptron discussed by Minsky-Papert '69. The paper further complements this result with a lower-bound on the generalization error of descent algorithms, which implies in particular that the robust universality breaks down if the gradients are averaged over large enough batches of samples as in full-GD, rather than fewer samples as in SGD.

## 1 Introduction

It is known that the class of neural networks (NNs) with polynomial network size can express any function that can be implemented in a given polynomial time [Par94, Sip06], and that their sample complexity scales polynomially with the network size [AB09]. Thus NNs have favorable approximation and estimation errors. However there is no known efficient training algorithm for NNs with general and provable guarantees, in particular, it is NP-hard to implement the ERM rule [KS09, DSS16]. The success behind deep learning is to train deep NNs with stochastic gradient descent or the like, which gives record performances in various applications [KSH12, HDY+12, LBBH98, LBH15, GBC16].

It is thus natural to ask whether SGD can also control efficiently the third pillar of statistical learning, i.e., the optimization error, turning deep learning into a universal learning paradigm that can learn efficiently any efficiently learnable class; see [SSBD14] for further discussions on this question.

This paper answers this question in the affirmative, with the following contributions and implications:

1. It is shown[1] that poly-size neural nets trained by SGD with poly-many steps can learn any function class that is learnable by an algorithm that runs in polytime and with poly-many samples; see Theorem 1. This part is resolved using a net initialization that is implemented in polytime (and not dependent on the function to be learned nor the data) and that emulates with SGD any efficient learning algorithm. This shows in particular that SGD-based deep learning is P-complete: any algorithm in P can be reduced to training with SGD a neural net initialized in polytime with a proper non-linearity and evaluating the net (see Remark 2).

2. We further show that this positive result is achieved with robustness: polynomial noise can be added to the gradients and weights can be of polynomial precision and the result still holds; see Theorem 2. Therefore, in a learning theoretic sense, deep learning gives a universal learning paradigm: approximation, estimation and also optimization errors are all controllable with polynomial parameters, and this is not degenerate as it can be implemented with polynomial precision. This also creates a *separation between deep learning and statistical query algorithms,* as the latter are not comparably universal due to cases like parities [Kea98].

3. Parities were known to be challenging since the work of Minsky-Papert for the perceptron [MP87], and our positive result requires indeed more than a single hidden layer to succeed, i.e., $O(\log n)$ layers[2] (see Example 1 in Appendix). In particular, our universality result together with [Raz16] imply that there exists function classes that *require large enough nets* to be learned with SGD: we know from [Raz16] that a net with $o(n^2/\log(n))$ edges of polynomial precisions cannot learn parities with poly-many samples, and thus with SGD, in polytime (even though one can represent parities with such size and depth 2); our result now shows that a net of size $n^2$ and $O(\log n)$ layers can learn parities with SGD in polytime.

4. A lower-bound is derived for descent algorithms on neural nets that shows that learning is impossible with polynomial precision if the *junk-flow* does not overcome the *cross-predictability*; see definitions in Section 2 and Theorem 3. The cross-predictability corresponds to an inverse average-case notion of statistical dimension that is classical in SQ algorithms [BFJ+94, Kea98, BKW03, Fel16], and the junk-flow is a quantity that is specific to descent algorithms. This shows in particular that the robust universality does not hold when replacing the stochastic gradients with perfect gradients on the entire population distribution or with large enough polynomial batches of fresh samples, in agreement with the results from SQ algorithms [Kea98, BFJ+94]. Therefore, some small amount of stochasticity[3] is needed to obtain the robust universality in our setting. The junk-flow also gives a measure for tackling lower-bounds for gradient descent algorithms more specifically.

In a practical setting, there may be no reason to use our SGD replacement to a general learning algorithm, but this universality result emphasizes the breadth of deep learning in the computational learning context and the fact that negative results about deep learning cannot be obtained without further constraints. A natural direction to pursue is typical architectures and initializations.

## 1.1 Problem formulations and learning objectives

We focus on Boolean functions to simplify the setting. Since it is known that any Boolean function that can be computed in time $O(T(n))$ can also be expressed by a neural network of size $O(T(n)^2)$ [Par94, Sip06], it is not meaningful to ask whether any such function $f_0$ can be learned with a poly-size NN and a descent algorithm that can depend on $f_0$; one can simply pre-set the net to express $f_0$. Two more meaningful questions are: (1) Can one learn a given function with an agnostic/random initialization? (2) Can one learn an unknown function from a class or distribution with a proper initialization?

For the second question, one is not given a specific function $f_0$ but a class of functions, or more generally, a distribution on functions. Therefore, one can no longer preset the net as desired in an obvious way. We focus here mainly on question 2, which is classical in statistical learning [SSBD14], and which gives a more general framework than restricting the initialization to be random. Moreover, in the case of symmetric function distributions, such as the parities discussed below, failure at 2 implies failure at 1. Namely, if we cannot learn a parity function for a random selection of the support $S$ (see definitions below), we cannot learn any given parity function on a typical support $S_0$ with a random initialization of the net, because the latter is symmetrical.

We thus have the following setting:

- Let $\mathcal{D} = \{+1, -1\}$ and $\mathcal{X} = \mathcal{D}^n$ be the data domain and let $\mathcal{Y} = \{+1, -1\}$ be the label domain. We work with binary vectors and binary labels for convenience (several of the results extend beyond this setting with appropriate reformulation of definitions).
- Let $P_{\mathcal{X}}$ be a probability distribution on the data domain $\mathcal{X}$ and $P_{\mathcal{F}}$ be a probability distribution on $\mathcal{Y}^{\mathcal{X}}$ (the set of functions from $\mathcal{X}$ to $\mathcal{Y}$). We also assume for convenience that these distributions lead to balanced classes, i.e., that $P(F(X) = 1) = 1/2 + o_n(1)$ when $(X, F) \sim P_{\mathcal{X}} \times P_{\mathcal{F}}$ (non-balanced cases require adjustments of the definitions).
- Our goal is to learn a function $F$ drawn under $P_{\mathcal{F}}$ by observing labelled examples $(X, Y)$ with $X \sim P_{\mathcal{X}}$, $Y = F(X)$.
- In order to learn $F$ we can train our algorithm on labelled examples with a descent algorithm starting with an initialization $f^{(0)}$ and running for a number of steps $T = T(n)$ (other parameters of the algorithm such as the learning rate are also specified). In the case of perfect GD, each step accesses the full distribution of labelled examples, while for SGD, it only accesses a single labelled example per step (see definitions below). In all cases, after the training with $(f^{(0)}, T)$, the algorithm produces an estimator $\hat{F}_{f^{(0)}, T}$ of $F$. We say that an algorithm achieves an accuracy of $\alpha$ in $T$ time steps for the considered $(P_{\mathcal{X}}, P_{\mathcal{F}})$, if a net with initialization $f^{(0)}$ can be constructed such that:

$$P(\hat{F}_{f^{(0)}, T}(X) = F(X)) \geq \alpha, \tag{1}$$

where the above probability is over $(X, F) \sim (P_{\mathcal{X}} \times P_{\mathcal{F}})$ and any randomness potentially used by the algorithm. We refer to typical-weak learning when $\alpha = 1/2 + \Omega_n(1)$. In other words, when we can predict the label of a new fresh sample from $P_{\mathcal{X}}$ with accuracy strictly better than random guessing.

Failing at typical-weak learning implies failing at most other learning requirements, such as PAC learning a class for the case of a uniform distribution on a certain class of functions [BKW03, MRT12]. For our positive results with SGD, we will not only show that one can efficiently typically weakly learn any function distribution that is efficiently typically weakly learnable, but that we can in fact reproduce whatever accuracy an algorithm can achieve for the considered distribution. We also shorten 'typical-weak learning' to simply 'learning' and talk about learning a 'function distribution' or a 'distribution' when referring to learning a pair $(P_{\mathcal{X}}, P_{\mathcal{F}})$.

**Example.** The problem of learning parities corresponds to $P_{\mathcal{X}}$ being uniform on $\{+1, -1\}^n$ and $P_{\mathcal{F}}$ uniform on the set of parity functions defined by $\mathcal{P} = \{p_s : s \subseteq [n]\}$, where $p_s : \{+1, -1\}^n \to \{+1, -1\}$ is such that $p_s(x) = \prod_{i \in s} x_i$. So nature picks $S$ uniformly at random, and with knowledge of $\mathcal{P}$ but not $S$, the problem is to learn which set $S$ was picked from samples $(X, p_S(X))$.

## 2    Results

### 2.1    Definitions and models

We use a fairly generic notion of neural nets, simply weighted directed acyclic graphs with a special vertex for the output, a special set of vertices for the inputs, and a non-linearity at the other vertices.

**Definition 1.** *A neural net is defined by a pair of a non-linearity function $f : \mathbb{R} \to \mathbb{R}$ and a weighted directed graph $G$ with some special vertices and the following properties. $G$ does not contain any cycle and there exists $n > 0$ such that $G$ has exactly $n + 1$ vertices that have no edges ending at them, $v_0, v_1, ..., v_n$. We refer to $n$ as the input size, $v_0$ as the constant vertex and $v_1, v_2, ..., v_n$ as the input vertices. Further, there exists a vertex $v_{out}$ such that for any other vertex $v'$, there is a path from $v'$ to $v_{out}$ in $G$. We also denote by $W = w(G)$ the weights on the edges of $G$. We denote by $eval_{(f,G)}(x)$ the evaluation of neural net $(f, G)$ at an input $x$ (or $eval_{(G)}(x)$ if $f$ is implicit). We also use the shortcut notation $W(x)$ for $\mathrm{eval}_G(x)$, when $G$ is implicit, with a slight abuse of notation between $W(G)$ and $W(X)$ (but the argument in $W()$ clarifies the definition).*

For a loss function $L$, a target function $h$, and a net $(f, G)$, the net's loss at a given input $x$ is $L(h(x) - eval_{(f,G)}(x))$.

Note that as we have defined them, neural nets generally give outputs in $\mathbb{R}$ rather than $\{0, 1\}$. As such, when talking about whether training a neural net by some method learns Boolean functions,

we will implicitly be assuming that the output of the net on the final input is thresholded at some predefined value or the like. None of our results depend on exactly how we deal with this part.

**Definition 2.** *Let $n > 0$, $\alpha \in [0,1]$, $P_{\mathcal{X}}$ be a probability distribution on $\{0,1\}^n$, and $P_{\mathcal{F}}$ be a probability distribution on the set of functions from $\{0,1\}^n$ to $\{0,1\}$. Also, let $X_0, X_1, ...$ be independently drawn from $P_{\mathcal{X}}$ and $F \sim P_{\mathcal{F}}$. An algorithm learns $(P_{\mathcal{F}}, P_{\mathcal{X}})$ with accuracy $\alpha$ in $T$ time steps if the algorithm is given the value of $(X_i, F(X_i))$ for each $i < T$ and, when given the value of $X_T \sim P_{\mathcal{X}}$ independent of $F$, it returns $Y_T$ such that $\mathbb{P}(F(X_T) = Y_T) \geq \alpha$.*

Algorithms such as SGD (or Gaussian elimination from samples) fit under this definition. For SGD, the algorithm starts with an initialization $W^{(0)}$ of the neural net weights, and updates it sequentially with each sample $(X_i, F(X_i))$ as $W^{(i)} = g(X_i, F(X_i), W^{(i-1)})$ where $g(X_i, F(X_i), W^{(i-1)}) = W^{(i-1)} - \gamma \nabla L(\text{eval}_{W^{(i-1)}}(X_i), F(X_i))$, $i < T$. It then outputs $Y_T = \text{eval}_{W^{(T-1)}}(X_T)$. For SGD with batch-size $m$ and fresh samples, one has to update the previous definition with not a single sample at each time step but $m$ i.i.d. samples at each time step, computing the empirical average of the query. The extreme case of perfect-GD corresponds to $m$ being infinity. So GD proceeds successively with the following $(F, P_{\mathcal{X}})$-dependent updates $W^{(i)} = \mathbb{E}_{X \sim P_{\mathcal{X}}} g(X, F(X), W^{(i-1)})$ for $i < T$ for the same function $g$ as in SGD. We also consider a noisy version of the above, to ensure that the algorithm is not succeeding due to infinite precision but with robustness.

## 2.2 Positive results

Our first result shows that for any distribution that can be learned by some algorithm in polytime, with poly-many samples and with accuracy $\alpha$, there exists an initialization (which means a neural net architecture with an initial assignment of the weights) that is constructed in polytime and that is agnostic to the function to be learned, such that training this neural net with SGD and possibly poly-noise learns this distribution in poly-steps with accuracy $\alpha - o(1)$.

**Theorem 1.** *For each $n > 0$, let $P_{\mathcal{X}}$ be a probability measure on $\{0,1\}^n$, and $P_{\mathcal{F}}$ be a probability measure on the set of functions from $\{0,1\}^n$ to $\{0,1\}$. Next, define $\alpha = \alpha_n$ such that there is some algorithm that takes a polynomial number of samples $(X_i, F(X_i))$ where the $X_i$ are i.i.d. under $P_{\mathcal{X}}$, runs in polynomial time, and learns $(P_{\mathcal{F}}, P_{\mathcal{X}})$ with accuracy $\alpha$. Then there exists $\gamma = o(1)$, a polynomial-sized neural net $(G_n, \phi)$ constructed in polytime, and a polynomial $T_n$ such that using stochastic gradient descent with learning rate $\gamma$ to train $(G_n, \phi)$ on $T_n$ samples $((X_i, R_i, R_i'), F(X_i))$ where[4] $(X_i, R_i, R_i') \sim P_{\mathcal{X}} \times \text{Ber}(1/2)^2$ learns $(P_{\mathcal{F}}, P_{\mathcal{X}})$ with accuracy $\alpha - o(1)$.*

**Remark 1.** *As a special case, one can construct in poly-time a net $(f, g)$ that has poly-size such that for a learning rate $\gamma$ and an integer $T$ that are at most polynomial, $(f, g)$ trained by SGD with learning rate $\gamma$ and $T$ time steps learns parities with accuracy $1 - o(1)$. In other words, random bits are not needed for parities, as parities can be learned with a deterministic algorithm using only samples of the same label without producing bias (see Section 4). Previous theorem also implies that SGD on neural nets can efficiently PAC-learn parities (or any class that is efficiently PAC-learnable).*

We now show that the previous result can be extended when sufficiently low amounts of inverse-polynomial noise are added to the weight of each edge in each time step. In particular, the previous theorem is not a degeneracy due to infinite precision.

**Theorem 2.** *For each $n > 0$, let $P_{\mathcal{X}}$ be a probability measure on $\{0,1\}^n$, and $P_{\mathcal{F}}$ be a probability measure on the set of functions from $\{0,1\}^n$ to $\{0,1\}$. Let $t_n$ polynomial in $n$. Next, define $\alpha = \alpha_n$ such that there is some algorithm that takes $t_n$ samples $(X_i, F(X_i))$ where the $X_i$ are independently drawn from $P_{\mathcal{X}}$ and $F \sim P_{\mathcal{F}}$, runs in polynomial time, and learns $(P_{\mathcal{F}}, P_{\mathcal{X}})$ with accuracy $\alpha$. Then there exists $\gamma = \Theta(1)$, and a polynomial-sized neural net $(G_n, f)$ such that using perturbed stochastic gradient descent with precision noise[5] $\delta \in [-1/(n^2 t_n), 1/(n^2 t_n)]^{t_n \times |E(G_n)|}$, learning rate $\gamma$, and loss function $L(x) = x^2$ to train $(G_n, f)$ on $t_n$ samples[6] $((X_i, R_i), F(X_i))$ where $(X_i, R_i) \sim P_{\mathcal{X}} \times \text{Ber}(1/2)$ learns $(P_{\mathcal{F}}, P_{\mathcal{X}})$ with accuracy $\alpha - o(1)$.*

**Corollary 1.** *For any $c > 0$, there exists a universal polytime initialization of a poly-size neural net, such that if samples are produced from a distribution that is learnable with accuracy $\alpha$ by some*

*algorithm working with an upper bound $n^c$ on the number of samples and the time needed per sample, then SGD run in polytime with poly-many samples and possibly inverse-poly noise will succeed in learning the distribution with accuracy $\alpha - o_n(1)$.*

**Remark 2.** *More generally, the process of training a neural net with noisy SGD is P-complete in the following sense. Let $A$ be a polynomial time algorithm that receives a binary string as input and then returns a value in $\{0,1\}$. For every $T$ polynomial in $n$ there exists a neural net $(G_n, f)$, learning rate and inverse polynomial level of noise such that when this net is trained for $T$ time steps on $(X_0, Y_0), ..., (X_{T-1}, Y_{T-1})$ using noisy SGD and then run on $X_T$ it returns $A(X_0, Y_0, X_1, Y_1, ..., Y_{T-1}, X_T)$ with high probability for all possible $(X_0, Y_0), ..., (X_{T-1}, Y_{T-1}), X_T$. The previous theorem is simply the case of this where $A$ is an algorithm that learns a function from random samples.*

**Remark 3.** *While the learning algorithm used in Theorem 2 does not put a bound on how large the edge weights can get during the learning process, we can do this in such a way that there is a constant that the weights will never exceed.*

Recall that in SQ algorithms [Kea98], for a query $\phi$, the oracle returns $\mathbb{E}_X \phi(X, F(X))$ within an error of $\tau$, where the range of $\phi$ is typically normalized to 1. Note that the expectation is on the population distribution, so SGD is not an SQ algorithm in that sense. In order for SQ to learn a function class, [BFJ$^+$94] shows that the number of queries times the signal-to-noise ratio $1/\sigma^2$ has to overcome the statistical dimension of the class. Since the statistical dimension of parities is exponentially large, under polynomially small noise, SQ algorithms cannot learn parities with polynomially many queries.

**Remark 4.** *Theorem 2 shows in particular that parities can be learned efficiently by SGD on neural nets (see Example 1 in Appendix for more details), even with an amount of noise that is polynomial and that prevents SQ algorithms from learning parities. Thus, Theorem 2 shows a separation between SGD-based deep learning and SQ algorithms. We further discuss this phenomenon in the next section.*

## 2.3 Negative results

### 2.3.1 GD and large averages

We saw that training neural nets with SGD and polynomial parameters is universal in that it can learn any efficiently learnable distribution. We now give a lower-bound for learning with a family of "descent algorithms" which includes GD and SGD. This implies in particular that the universality is lost once perfect gradients are used, or once a large number of fresh samples are used to average each gradient, in agreement with the bounds from SQ algorithms [BFJ$^+$94, FGV17, Kea98, BKW03, Fel16, Yan05, FGR$^+$17, SVW15]. The theorem also gives a new quantity, the "junk-flow", which can be used to lower bound the performance of "descent algorithms" beyond the number of queries.

**Definition 3** (Descent algorithms). *Consider for each $n > 0$ a neural net of size $|E(n)|$ initialized with weights $W^{(0)}$. A descent algorithm running for $T$ time steps is defined by a sequence of query functions $\{G_t\}_{t \in [T]}$ that rely at each time steps on $m$ samples[7], a query range[8] of $A$, a parameter $\sigma^2$ for the noise variance, and operates by updating at each iterate the weights by*

$$W^{(t)} = W^{(t-1)} - \mathbb{E}_{X \sim \hat{P}_{S_m^{(t)}}} G_{t-1}(W^{(t-1)}(X), F(X)) + Z^{(t)}, \quad t = 1, \ldots, T \qquad (2)$$

*where $\{Z^{(t)}\}_{t \in [T]}$ are i.i.d. $\mathcal{N}(0, \sigma^2 I_{|E(n)|})$, $\{S_m^{(t)}\}_{t \in [T]}$ are i.i.d. with $S_m^{(t)} = (X_1^{(t)}, \ldots, X_m^{(t)})$ i.i.d. under $P_{\mathcal{X}}$ ($\hat{P}_{S_m^{(t)}}$ denotes the empirical distribution of $S_m^{(t)}$), and $\{Z^{(t)}\}_{t \in [T]}$ and $\{S_m^{(t)}\}_{t \in [T]}$ are independent.*

We use the notation $S_m^{(<t)} = (S_m^{(1)}, \ldots, S_m^{(t-1)})$ and $S_m^{(\leq t)} = (S_m^{(1)}, \ldots, S_m^{(t)})$.

**Remark 5.** *Note if one constraints the net architecture and initialization to be classical ones, then a descent algorithm is more restrictive than an SQ algorithm because it forces the algorithm to make edits on the memory (i.e., the neural net) by making sequential linear corrections as in (2), whereas*

*SQ algorithms can store and adapt the queries as desired. Note also that for a differentiable loss L, $G_t = \gamma_t[\nabla L]_A$ gives gradient descent with fresh batches, and we refer to the case where X is drawn from the true data distribution, i.e, $m = \infty$ as perfect-GD.*

**Definition 4** (junk-flow). *Using the notation in the previous definition, define the junk flow of an initialization $W^{(0)}$ with data distribution $P_{\mathcal{X}}$, T steps and queries $\{G_t\}_{t \in [T]}$ by*

$$\text{JF} = \text{JF}(W^{(0)}, P_{\mathcal{X}}, \{G_t\}_{t \in [T]}) := \sum_{t=1}^{T}(\mathbb{E}_t\|G_t(W_{\star}^{(t-1)}(X), Y)\|_2^2)^{1/2}. \tag{3}$$

*where $(X, Y) \sim P_{\mathcal{X}} \times U_{\mathcal{Y}}$ are independent of all other random variables, $W_{\star}^{(0)} = W^{(0)}$, $W_{\star}^{(t)} = W_{\star}^{(t-1)} - (1/m)\sum_{i=1}^{m} G_t(W_{\star}^{(t-1)}(X_i^{(t)}), Y_i^{(t)}) + Z^{(t)}$, $t \in [T]$, with $Y_i^{(t)}$ i.i.d. under $U_{\mathcal{Y}}$ and independent of all other random variables, and $\mathbb{E}_t$ is the expectation over $S_m^{(<t)}, Z^{(<t)}, X, Y$. That is, the junk-flow is the power series over all time steps of the root of the expected gradient squared-norm when running GD on random samples with completely random labels.*

**Definition 5** (Cross-predictability). *For a positive integer m, a probability measure $P_{\mathcal{X}}$ on the data domain $\mathcal{X}$, and a probability measure $P_{\mathcal{F}}$ on the class of functions $\mathcal{F}$ from $\mathcal{X}$ to $\mathcal{Y} = \{+1, -1\}$, we define the cross-predictability (of order m) by*

$$\text{CP}_m(P_{\mathcal{X}}, P_{\mathcal{F}}) := \mathbb{E}_{(X^m, F, F') \sim P_{\mathcal{X}}^m \times P_{\mathcal{F}} \times P_{\mathcal{F}}}(\mathbb{E}_{X \sim P_{X^m}} F(X)F'(X))^2, \tag{4}$$

*where $X^m = (X_1, \ldots, X_m)$ has i.i.d. components under $P_{\mathcal{X}}$, $F, F'$ are independent of $X^m$ and i.i.d. under $P_{\mathcal{F}}$, and X is drawn independently of $(F, F')$ under the empirical measure of $X^m$, i.e., $P_{X^m} = \frac{1}{m}\sum_{i=1}^{m}\delta_{X_i}$.*

Note the following equivalent representations:

$$\text{CP}_m(P_{\mathcal{X}}, P_{\mathcal{F}}) = \frac{1}{m} + \left(1 - \frac{1}{m}\right)\text{CP}_{\infty}(P_{\mathcal{X}}, P_{\mathcal{F}}), \tag{5}$$

$$\text{CP}_{\infty}(P_{\mathcal{X}}, P_{\mathcal{F}}) := \mathbb{E}_{F, F' \sim P_{\mathcal{F}}}(\mathbb{E}_{X \sim P_{\mathcal{X}}} F(X)F'(X))^2 \tag{6}$$

$$= \mathbb{E}_{X, X' \sim P_{\mathcal{X}}}(\mathbb{E}_{F \sim P_{\mathcal{F}}} F(X)F(X'))^2 = \|\mathbb{E}_F \mathcal{F}(F)^{\otimes 2}\|_2^2 \tag{7}$$

where $\mathcal{F}(F)$ denotes the Fourier-Walsh transform of F with respect to the measure $P_{\mathcal{X}}$.

This measures how predictable a sampled function is from another one on a typical data point, or equivalently, how predictable a sampled data label is from another one on a typical function. Equivalently, this measures the typical correlation among functions, similarly to the average statistical dimension [FGR+17, FPV18]. Note that the data point is drawn from the empirical distribution on m samples, where m will refer to the batch-size of a set of fresh samples in the context of GD (i.e., how many samples are used to compute gradients). For $m = \infty$, i.e., perfect statistics, we have for example that if $P_{\mathcal{X}}$ is a delta function, $\text{CP}_{\infty}$ achieves the largest possible value of 1, and for purely random input and purely random functions, $\text{CP}_{\infty}$ is $2^{-n}$, the lowest possible value. For random degree-k monomials and uniform inputs, $CP_{\infty} \asymp \binom{n}{k}^{-1}$. We now present the lower-bound.

**Theorem 3.** *Let $P_{\mathcal{X}}$ with $\mathcal{X} = \mathcal{D}^n$ for some finite set $\mathcal{D}$ and $P_{\mathcal{F}}$ such that the output distribution is balanced,[9] i.e., $\mathbb{P}\{F(X) = 0\} = \mathbb{P}\{F(X) = 1\} + o_n(1)$ when $(X, F) \sim P_{\mathcal{X}} \times P_{\mathcal{F}}$. Using the previous definitions for $\text{CP}_m = CP_m(P_{\mathcal{X}}, P_{\mathcal{F}})$ and $\text{JF}_T = \text{JF}(W^{(0)}, P_{\mathcal{X}}, \{G_t\}_{t \in [T]})$, the generalization error of a descent algorithm as in Definition 3 is lower-bounded as*

$$\mathbb{P}\{W^{(T)}(X) \neq F(X)\} \geq 1/2 - \frac{1}{\sigma} \cdot \text{JF}_T \cdot \text{CP}_m^{1/4} \tag{8}$$

$$\geq 1/2 - \frac{1}{\sigma} \cdot \text{JF}_T \cdot (1/m + \text{CP}_{\infty})^{1/4}. \tag{9}$$

In Section 5 of the Appendix, we present a stronger version of Theorem 3 for parities (Theorem 6), with a tighter bound obtained that results in the term $\text{CP}^{1/2}$ rather than $\text{CP}^{1/4}$.

**Corollary 2.** *We have* $\mathrm{JF}_T \leq T\sqrt{|E|}A$, *and a descent algorithm as in previous theorem with* $M := \max(\frac{1}{\sigma}, A, |E|, T)$ *polynomial in $n$ cannot learn under $(P_\mathcal{X}, P_\mathcal{F})$ if $\mathrm{CP}_m$ decays super-polynomially in $n$ (or more precisely if $1/\mathrm{CP}_m$ is a large enough polynomial). In particular, a descent algorithm as in previous theorem and $\max(\frac{1}{\sigma}, A, |E|, T)$ polynomial in $n$ can learn a random degree-$k$ monomial with perfect-GD (or $m$ super-polynomial) if and only if* [10] *$k = O(1)$.*

**Remark 6.** *The above corollary gives a bound similar to those that can be obtained using SQ algorithms. Various results from SQ need to be combined to obtain comparable bounds. One needs to account for the statistical nature of the noise that we consider here, which has less degrees of freedom compared to the adversarial noise of standard SQ, and one needs to account for the weak and typical learning requirements (equ. 1). These can be addressed - at least in the case where the distribution $P_\mathcal{F}$ is uniform on a set - using concentration and coupling arguments and combining for example results from [Fel16, Szö09, FPV18]. We refer to [Boi19] for a detailed discussion on how this is done, which gives the following bound $\mathbb{P}\{W^{(T)}(X) \neq F(X)\} \geq \frac{1}{2} - O(\frac{1}{\sigma^{2/3}} A^{2/3} \Pi^{1/12} |E|T)$ in the context of the descent algorithms considered here. This can be further improved to $\mathbb{P}\{W^{(T)}(X) \neq F(X)\} \geq \frac{1}{2} - O(\frac{1}{\sigma^{1/2}} A^{1/2} \Pi_1^{1/4} |E|T)$ by using an $L_1$-notion of cross-predictability $\Pi$ (replacing the square of the inner-product by the absolute value). Such bounds are slightly weaker than the one of Corollary 2. Note that these use a coupling argument to handle the adversarial v.s. statistical noise, whereas if the latter were considered, the bound would turn to $\mathbb{P}\{W^{(T)}(X) \neq F(X)\} \geq \frac{1}{2} - O(\frac{1}{\sigma^2} A^2 \Pi^{1/2} |E|T + \Pi^{1/4})$ and $\frac{1}{2} - O(\frac{1}{\sigma} A \Pi_1^{1/2} |E|T)$ with the $L_1$-cross-predictability (note that we can also obtain an exponent of $1/2$ on the cross-predictability with our Theorem 6 in the Appendix). Further improvements may be obtained using [Yan05] for the statistical noise, but all together these bounds are of a similar kind. We next discuss how the junk-flow could lead to different kinds of bounds in the context of neural networks.*

**Remark 7.** *The upper bound on the junk flow Corollary 2 uses a simple upper bound on the derivative of the loss function. In cases where the derivatives of the output with respect to the edge weights will consistently be much smaller than $A$, one can prove tighter bounds on the junk flow, leading to a lower probability of learning the function. One could also obtain comparable improvements in the SQ bounds by adjusting the SQ algorithm to take these bounds on the derivatives into account. However, in cases where there are some inputs for which the derivative of the output with respect to the edge weights are much larger than for typical inputs, one could obtain tighter bounds on the junk flow that do not necessarily have analogous tighter SQ bounds. We leave this to future work.*

**Remark 8.** *Note that our positive results show that we could learn a random parity function using stochastic gradient descent. The difference is that SGD lets us get the details of single samples (the stochastic gradients do not necessarily cancel out), which is needed (for example) to come up with algorithms like Gaussian elimination for parities. If instead one uses GD on the entire population, the average of all possible samples mostly cancels out in cases like parities, so that an exponentially small amount of noise is enough to drown out whatever signal is left in the gradient (which becomes almost independent of the function). This does not take place if GD is used with a sub-polynomial batch-size, or for functions that correlate well with the neural nets.*

## 2.4 Proof techniques

**Positive results.** For the positive results, we emulate any learning algorithm using poly-many samples and running in poly-time with poly-size neural nets trained by poly-step SGD. This requires emulating any poly-size circuit implementation with free access to reading and writing in memory using a particular computational model that computes, reads and writes memory solely via SGD steps on a fixed neural net. In particular, this requires designing subnets that perform arbitrary efficient computations in such a way that SGD does not alter them and subnet structures that cause SGD to change specific edge weights in a manner that we can control.

Note that any algorithm that learns a function from samples must repeatedly get a new sample and then change some of the values in its memory in a way that is determined by the current values in its memory and the value of the sample. Eventually, it must also attempt to compute the function's output based on its input and the values in memory. If the learning algorithm is efficient, then there must be a polynomial-sized circuit that computes the values in the algorithm's memory in the next timestep

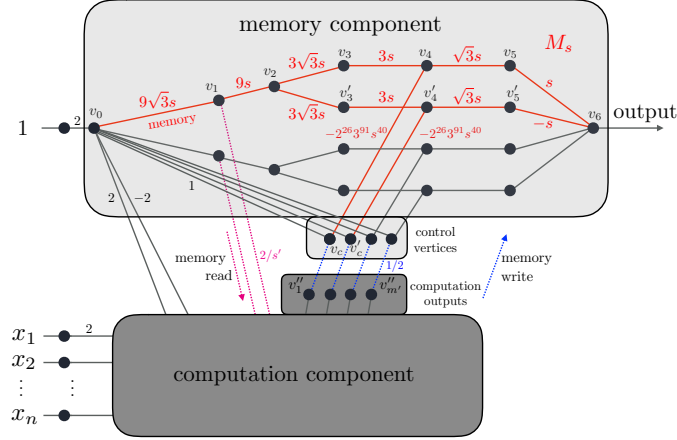

Figure 1: The emulation net; $s = \sqrt[364]{2^{-243}3^{-1641/2}/m'}$, $m'$ is the maximum between $m$ and $\lceil 2^{-243}3^{-1641/2}(18\sqrt{3})^{364}\rceil$, $m$ is the number of bits required to perform the computation from the computation component and $s' = (18\sqrt{3}s)^3$. We highlight in red (top two paths) one copy of $M_s$.

from the sample it was given and its memory values in the current timestep. Likewise, there must be a polynomial-sized circuit that computes its guesses of the function's output from the function's input and the values in its memory. Since any polynomial-sized circuit can be translated into a neural net of polynomial size, we can encode the desired circuit in a preset format. However, once we run SGD on it, we would a priori completely alter the weights of the edges in the net, which would cause the net to stop performing the intended calculations. To prevent this, we use an activation function that is constant in some areas (we will use a sigmoid like non-linearity with flats), and ensure that the nodes in the translated circuit are **flat nodes**, i.e., that they always get inputs in that flat range. That way, the derivatives of their activation levels with respect to the weights of any of the edges leading to them are 0, so backpropagation will never change the edge weights in the net. That allows us to construct a portion of the net called the **computation component** that performs the desired computations in a **backpropagation-proofed** way. This computation component can in particular output values, the **computation outputs**, that are responsible for editing the memory of the algorithm. However, we still need a mechanism to decide how to store and edit the memory using only a neural net trained with SGD. This is the most challenging part.

The neural net's memory takes the form of its edge weights. We will encode the algorithm's memory in the edges leaving the constant vertex. Normally, we would not be able to precisely control how SGD would alter these weights. However, it is possible to design components of the net (the $M_s$ components in Figure 2 defined in Definition 6 of Appendix) in such a way that if certain vertices called the **control vertices** output certain values, then every path to the output through a designated edge will pass through a flat vertex. So, if those vertices are set that way, the derivative of the loss function with respect to the edge weight in question will be 0, and the weight will not change. That allows us to control whether or not the edge weight changes, and by appropriately setting up the values of the initial net and the learning rate, we can ensure that the changes will always translate into the desired bit flips. This gives us a way to construct a net portion that can set the values in memory; we call this the **memory component**. See Figure 2 for a representation of the overall net.

One difficulty encountered with such an SGD implementation is that no update of the weights will take place when given a sample that is correctly predicted by the net. If one does not mitigate this, the net may end up being trained on a sample distribution that is mismatched to the original one, which can have unexpected consequences. A randomization mechanism is thus used to circumvent this issue, but this mechanism is not necessary for symmetric functions like parities, as one can learn parities only from samples having the same label.

In summary, we can create a neural net with a poly-size architecture and a poly-time initialization, that carries out when trained by SGD both the computation and memory updates of an algorithm. See the Appendix for additional implementation considerations.

**Negative results.** Our main approach to showing the failure of an algorithm (e.g., noisy GD) using data from a model (e.g, parities) for a desired task (e.g., typical weak learning), will be to show that under limited resources (e.g., limited number of time steps), the output of the algorithm trained on the true model is *statistically indistinguishable* from the output of the algorithm trained on a null model, where the null model fails to provide the desired performance for trivial reasons. This gives a computational lower-bound out of a statistical estimate. The indistinguishability to null condition (INC) is obtained by manipulating information measures, bounding the total variation distance of the two posterior measures between the test and null models.

More specifically, we show a subadditivity property of the TV using the data processing inequality, use the fact that we work with a descent algorithms that updates the weights by 'subtractions' of queries and not general statistical queries, bound the one step total variation distance with the KL distance (Pinsker's inequality), which in the Gaussian case gives the $\ell_2$ distance of the means. Then we use a change of measure argument, manipulating the Radon-Nikodym derivative with a tensorization argument to linearize the expression, and concluding with generic inequalities to bring up the junk-flow and the cross-predictability (using replicate random variables).

## 3    Further related literature

In [CRW18] a different emulation argument is shown for gradient descent, also encoding a calculation using a form of GD, but in very different settings and with very different conclusions. [CRW18] shows that one can implement an arbitrary algorithm using GD by repeating the correct series of loss functions, with the purpose of showing that it is difficult to predict the long-term results of running online GD on an arbitrary known series of loss functions. Our emulation shows that one can encode an arbitrary computation on *samples* drawn from an unknown distribution by training a net with *SGD*. Our purpose is to prove that a properly initialized net trained by SGD can learn any function learnable from samples. Finally, a key component of our result is that SGD can handle an amount of *noise* that goes beyond what SQ algorithms can handle, which is unrelated to [CRW18].

The difficulty of learning functions like parities with NNs is not new. Together with the connectivity case, the difficulty was one of the central foci in the perceptron book of Minksy and Papert [MP87] The sensitivity of parities is also well-studied in the theoretical computer science literature, with the relation to circuit complexity, in particular the computational limitations of small-depth circuits [Hås87, All96]. The seminal paper of Kearns on statistical query learning algorithms [Kea98] brings up the difficulties in learning parities with such algorithms. As mentioned earlier there have been numerous works extending the work of Kearns for parities to more general cases of high statistical dimension, such as [BFJ$^+$94, FGV17, Kea98, BKW03, Fel16, Yan05, FGR$^+$17, SVW15, Szö09, FPV18] and [SVWX17, VW18] for specific neural networks. While the statistical dimension was initially derived with a worst-case requirement on the class of functions, it was generalized to average-case notions in [FGR$^+$17, FPV18, Fel16] and statistical noise [Yan05]. Information measure manipulations as used in our lower-bound were also used in [SVW15] to obtain SQ bounds under memory constraints. We refer to [Boi19] for further comparisons on SQ algorithms. Finally, [SSS17], with an earlier version in [Sha18] from the first author, also investigates the impossibility of learning parities. In particular, [SSS17] proves that the gradient of the loss function of a neural network will be essentially independent of the parity function used, which gives strong indications for the failure of GD. This is achieved in [SSS17] under the requirement that the loss function is 1-Lipschitz, an assumption that is not needed in our Lemma 1 in the Appendix.

## Conclusions and directions

For free poly-time initializations, we proved that SGD on neural nets with polynomial parameters is as general as poly-time PAC learning, but that GD is limited as SQ. It would be interesting to further investigate our lower-bound in the context of random initializations, in particular the behavior of the junk-flow, and whether parities can still be learnt by SGD with 'more random' initializations. We also conjecture that for several architectures (e.g., fully connected layers) and random initializations (e.g., i.i.d. centered Gaussian weights of variance inverse proportional to the width), SGD/GD will not learn on a polynomial horizon a target function $f$ that has a net-to-target cross predictability that is negligible, i.e., $\max_v \mathbb{E}_{W^{(0)}} \langle f, f_{W^{(0)}}^{(v)} \rangle^2 = n^{-\omega(1)}$, where $f_{W^{(0)}}^{(v)}$ is the output of neuron $v$ in the net at initialization. In particular, this should hold for functions having no constant-degree monomials.

## Acknowledgments and Disclosure of Funding

Part of this work was supported by NSF CAREER Award CCF-1552131.

## Broader Impact

We do not anticipate any ethical aspects and future societal consequences due to the theoretical focus of this work.

## Footnotes

[1]This paper discusses results that are asymptotic in the dimension $n$ of the data, i.e., the number of inputs to the neural net. The term 'poly' is used to emphasize that the considered quantities scale polynomially with $n$, i.e., $n^c$ for a constant $c$. This can be polynomially large if $c \geq 0$ or polynomially small if $c < 0$. In some parts, we make abuse of terminologies and simply say poly/polynomially when it is clear from the context which direction is considered; e.g., only a noise variance that is polynomially small is interesing in our context, not a noise variance that is polynomially large (which would be a huge noise).

[2]One can reduce the number of layers by using threshold gates in the computation component of arbitrary fan-in; see Section 4.

[3]The stochasticity of SGD has also been advocated in contexts such as stability, implicit regularization or to avoid bad critical points [HRS16, ZBH+16, PP17, KLY18].

[4] We use $\text{Ber}(1/2)$ to denote the Bernoulli distribution on $\{0,1\}$.

[5] This means that each weight at each time step of SGD is perturbed by an amount bounded by $1/(n^2 t_n)$.

[6] Formally the samples should be converted to $((2X_i - 1, 2R_i - 1), 2F(X_i) - 1)$, i.e., valued in $\{-1, 1\}$ to be consistent with the rest of the notations.

[7] These are i.i.d. from the distribution $P_{\mathcal{X}}$ with labels from $F$ in the context of this section.

[8] We call the range of a function to be $A$ if any value of the function potentially exceeding $A$ (or $-A$) is rounded at $A$ (or $-A$). I.e., as for SQ, the signal magnitude cannot grow unbounded compared to the noise.

[9]Non-balanced cases can be handled by modifying definitions appropriately.

[10]The positive statement uses the fact that it is easy to learn random degree-$k$ monomials when $k$ is finite; see for example [Bam19] for a specific implementation.

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
