[Supplementary Material]

# 4 Proofs of positive results: universality of deep learning

## 4.1 Emulation of arbitrary algorithms

Any algorithm that learns a function from samples must repeatedly get a new sample and then change some of the values in its memory in a way that is determined by the current values in its memory and the value of the sample. Eventually, it must also attempt to compute the function's output based on its input and the values in memory. If the learning algorithm is efficient, then there must be a polynomial-sized circuit that computes the values in the algorithm's memory in the next timestep from the sample it was given and its memory values in the current timestep. Likewise, there must be a polynomial-sized circuit that computes its guesses of the function's output from the function's input and the values in its memory.

Any polynomial-sized circuit can be translated into a neural net of polynomial size. Normally, stochastic gradient descent would tend to alter the weights of edges in that net, which might cause it to stop performing the calculations that we want. However, we can prevent its edge weights from changing by using an activation function that is constant in some areas, and ensuring that the nodes in the translated circuit always get inputs in that range. That way, the derivatives of their activation levels with respect to the weights of any of the edges leading to them are $0$, so backpropagation will never change the edge weights in the net. That leaves the issue of giving the net some memory that it can read and write. A neural net's memory takes the form of its edge weights. Normally, we would not be able to precisely control how stochastic gradient descent would alter these weights. However, it is possible to design the net in such a way that if certain vertices output certain values, then every path to the output through a designated edge will pass through a vertex that has a total input in one of the flat parts of the activation function. So, if those vertices are set that way the derivative of the loss function with respect to the edge weight in question will be $0$, and the weight will not change. That would allow us to control whether or not the edge weight changes, which gives us a way of setting the values in memory. As such, we can create a neural net that carries out this algorithm when it is trained by means of stochastic gradient descent with appropriate samples and learning rate. This net will contain the following components:

1. *The output vertex.* This is the output vertex of the net, and the net will be designed in such a way that it always has a value of $\pm 1$.

2. *The input bits.* These will include the regular input vertices for the function in question. However, there will also be a couple of extra input bits that are to be set randomly in each timestep. They will provide a source of randomness that is necessary for the net to run randomized algorithms[11], in addition to some other guesswork that will turn out to be necessary (see more on this below).

3. *The memory component.* For each bit of memory that the original algorithm uses, the net will have a vertex with an edge from the constant vertex that will be set to either a positive or negative value depending on whether that bit is currently set to $0$ or $1$. Each such vertex will also have an edge leading to another vertex which is connected to the output vertex by two paths. The middle vertex in each of these paths will also have an edge from a control vertex. If the control vertex has a value of $2$, then that vertex's activation will be $0$, which will result in all subsequent vertices on that path outputting $0$, and none of the edge weights on that path changing as a result of backpropagation along that path. On the other hand, if the control vertex has a value of $0$, then that vertex will have a nonzero activation, and so will all subsequent vertices on that path. The learning rate will be chosen so that in this case, if the net gives the wrong output, the weight of every edge on this path will be multiplied by $-1$. This will allow the computation component to set values in memory using the control vertices. (See definition 6 and lemma 2 for details on the memory component.)

4. *The computation component.* This component will have edges leading to it from the inputs and from the memory component. It will use the inputs and the values in memory to compute what the net should output and what to set the memory bits to at the end of the current timestep if the net's output is wrong. There will be edges leading from the appropriate vertices in this component to the control vertices in the memory component in order to set

Figure 2: The emulation net. The parameters are $s = \sqrt[364]{2^{-243}3^{-1641/2}/m'}$, where $m' = \max(m, \lceil 2^{-243}3^{-1641/2}(18\sqrt{3})^{364}\rceil)$, $s' = (18\sqrt{3}s)^3$ and $m$ is the total number of bits required to perform the computation from the computation component. In this illustration, we considered only two copies of the $M_s$ from Definition 6; one copy is highlighted in red. The magenta dashed edges are the memory read edges and the blue dashed edges are the memory write edges. The latter allow to change the controller vertices $v_c, v_c'$ that act on $M_s$ to edit the memory. Random bit inputs are omitted in this figure, and the information flows from left to right in all edges.

the bits to the values it has computed. If the net's output is right, the derivative of the loss function with respect to any edge weight will be $0$, so the entire net will not change. This component will be constructed in such a way that the derivative of the loss function with respect to the weights of its edges will always be $0$. As a result, none of the edge weights in the computation component will ever change, as explained in lemma 1. This component will also decide whether or not the net has learned enough about the function in question based on the values in memory. If it thinks that it still needs to learn, then it will have the net output a random value and attempt to set the values in memory to whatever they should be set to if that guess is wrong. If it thinks that it has learned enough, then it will try to get the output right and leave the values in memory unchanged.

See Figure 2 for a representation of the overall net. One complication that this approach encounters is that if the net outputs the correct value, then the derivative of the loss function with respect to any edge weight is $0$, so the net cannot learn from that sample.[12] Our approach to dealing with that is to have a learning phase where we guess the output randomly and then have the net output the opposite of our guess. That way, if the guess is right the net learns from that sample, and if it is wrong it stays unchanged. Each guess is right with probability $1/2$ regardless of the sample, so the probability distribution of the samples it is actually learning from is the same as the probability distribution of the samples overall, and it only needs $(2 + o(1))$ times as many samples as the original algorithm in order to learn the function. Once it thinks it has learned enough, such as after learning from a

designated number of samples, it can switch to attempting to compute the function it has learned on each new input.

**Example 1.** *We now give an illustration of how previous components would run and interact for learning parities. One can learn an unknown parity function by collecting samples until one has a set that spans the space of possible inputs, at which point one can compute the function by expressing any new input as a linear combination of those inputs and returning the corresponding linear combination of their outputs. As such, if we wanted to design a neural net to learn a parity function this way, the memory component would have $n(n+1)$ bits designated for remembering samples, and $\log_2(n+1)$ bits to keep a count of the samples it had already memorized. Whenever it received a new input $x$, the computation component would get the value of $x$ from the input nodes and the samples it had previously memorized, $(x_1, y_1), ..., (x_r, y_r)$, from the memory component. Then it would check whether or not $x$ could be expressed as a linear combination of $x_1, ..., x_r$. If it could be, then the computation component would compute the corresponding linear combination of $y_1, ..., y_r$ and have the net return it. Otherwise, the computation component would take a random value that it got from one of the extra input nodes, $y'$. Then, it would attempt to have the memory component add $(x, y')$ to its list of memorized samples and have the net return $NOT(y')$. That way, if the correct output was $y'$, then the net would return the wrong value and the edge weights would update in a way that added the sample to the net's memory. If the correct output was $NOT(y')$, then the net would return the right value, and none of the edge weights would change. As a result, it would need about $2n$ samples before it succeeded at memorizing a list that spanned the space of all possible inputs, at which point it would return the correct outputs for any subsequent inputs. Note that the depth of the emulation net is at least 7 in general due to the memory component, and here for parities, it can be done with depth $O(\log n)$ as the computation component can be implemented with this depth.[13]*

Before we can prove anything about how our net learns, we will need to establish some properties of our activation function. Throughout this section, we will use an activation function $f : \mathbb{R} \to \mathbb{R}$ such that $f(x) = 2$ for all $x > 3/2$, $f(x) = -2$ for all $x < -3/2$, and $f(x) = x^3$ for all $-1 < x < 1$. There is a way to define $f$ on $[-3/2, -1] \cup [1, 3/2]$ such that $f$ is smooth and nondecreasing. The details of how this is done will not affect any of our arguments, so we pick some assignment of values to $f$ on these intervals with these properties. This activation function has the important property that its derivative is $0$ everywhere outside of $[-3/2, 3/2]$. As a result, if we use SGD to train a neural net using this activation function, then in any given time step, the weights of the edges leading to any vertex that had a total input that is not in $[-3/2, 3/2]$ will not change. This allows us to create sections of the net that perform a desired computation without ever changing. In particular, it will allow us to construct the net's computation component in such a way that it will perform the necessary computations without ever getting altered by SGD. More formally, we have the following.

**Lemma 1** (Backpropagation-proofed circuit emulation). *Let $h : \{0, 1\}^m \to \{0, 1\}^{m'}$ be a function that can be computed by a circuit made of AND, OR, and NOT gates with a total of $b$ gates. Also, consider a neural net with $m$ input[14] vertices $v'_1, ..., v'_m$, and a collection of chosen real numbers $y_1^{(0)} < y_1^{(1)}, y_2^{(0)} < y_2^{(1)}, ..., y_m^{(0)} < y_m^{(1)}$. It is possible to add a set of at most $b$ new vertices to the net, including output vertices $v''_1, ..., v''_{m'}$, along with edges leading to them such that for any possible addition of edges leading from the new vertices to old vertices, if the net is trained by SGD and the output of $v'_i$ is either $y_i^{(0)}$ or $y_i^{(1)}$ for every $i$ in every timestep, then the following hold:*

1. *None of the weights of the edges leading to the new vertices ever change, and no paths through the new vertices contribute to the derivative of the loss function with respect to edges leading to the $v'_i$.*

2. *In any given time step, if the output of $v'_i$ encodes $x_i$ with $y_i^{(0)}$ and $y_i^{(1)}$ representing $0$ and $1$ respectively for each $i$[15], then the output of $v''_j$ encodes $h_j(x_1, ..., x_m)$ for each $j$ with $-2$ and $2$ encoding $0$ and $1$ respectively.*

*Proof.* In order to do this, add one new vertex for each gate in a circuit that computes $h$. When the new vertices are used to compute $h$, we want each vertex to output $2$ if the corresponding gate outputs a $1$ and $-2$ if the corresponding gate outputs a $0$. In order to make one new vertex compute the NOT of another new vertex, it suffices to have an edge of weight $-1$ to the vertex computing the NOT and no other edges to that vertex. We can compute an AND of two new vertices by having a vertex with two edges of weight $1$ from these vertices and an edge of weight $-2$ from the constant vertex. Similarly, we can compute an OR of two new vertices by having a vertex with two edges of weight $1$ from these vertices and an edge of weight $2$ from the constant vertex. For vertices corresponding to gates that act directly on the inputs, we have the complication that their vertices do not necessarily encode $0$ and $1$ as $\pm 2$, but we can compensate for that by changing the weights of the edges from these vertices, and the edges to these gates from the constant vertices appropriately.

This ensures that if the outputs of the $v_i'$ encode binary values $x_1, ..., x_m$ appropriately, then each of the new vertices will output the value corresponding to the output of the appropriate gate. So, these vertices compute $h(x_1, ..., x_m)$ correctly. Furthermore, since the input to each of these vertices is outside of $[-3/2, 3/2]$, the derivatives of their activation functions with respect to their inputs are all $0$. As such, none of the weights of the edges leading to them ever change, and paths through them do not contribute to changes in the weights of edges leading to the $v_i'$. $\qquad\square$

Note that any efficient learning algorithm will have a polynomial number of bits of memory. In each time step, it might compute an output from its memory and sample input, and it will compute which memory values it should change based on its memory, sample input, and sample output. All of these computations must be performable in polynomial time, so there is a polynomial sized circuit that performs them. Therefore, by the lemma it is possible to add a polynomial sized component to any neural net that performs these calculations, and as long as the inputs to this component always take on values corresponding to $0$ or $1$, backpropagation will never alter the weights of the edges in this component. That leaves the issue of how the neural net can encode and update memory bits. Our plan for this is to add in a vertex for each memory bit that has an edge with a weight encoding the bit leading to it from a constant bit and no other edges leading to it. We will also add in paths from these vertices to the output that are designed to allow us to control how backpropagation alters the weights of the edges leading to the memory vertices. More precisely, we define the following.

**Definition 6.** *For any positive real number $s$, let $M_s$ be the weighted directed graph with 12 vertices, $v_0$, $v_1$, $v_2$, $v_3$, $v_4$, $v_5$, $v_c$, $v_3'$, $v_4'$, $v_5'$, $v_c'$, and $v_6$ and the following edges:*

1. *An edge of weight $3^{3-t/2}s$ from $v_{t-1}$ to $v_t$ for each $0 < t \leq 6$*

2. *An edge of weight $3\sqrt{3}s$ from $v_2$ to $v_3'$*

3. *An edge of weight $3^{3-t/2}s$ from $v_{t-1}'$ to $v_t'$ for each $3 < t < 6$*

4. *An edge of weight $-s$ from $v_5'$ to $v_6$*

5. *An edge of weight $-2^{26} \cdot 3^{91} s^{40}$ from $v_c$ to $v_4$.*

6. *An edge of weight $-2^{26} \cdot 3^{91} s^{40}$ from $v_c'$ to $v_4'$.*

We refer to Figure 2 to visualize $M_s$. The idea is that this structure can be used to remember one bit, which is encoded in the current weight of the edge from $v_0$ to $v_1$. A weight of $9\sqrt{3}s$ encodes a $0$ and a weight of $-9\sqrt{3}s$ encodes a $1$. In order to set the value of this bit, we will use $v_c$ and $v_c'$, which will be controlled by the computation component. If we want to keep the bit the same, then we will have them both output $2$, in which case $v_4$ and $v_4'$ will both output $0$, with the result that the derivative of the loss function with respect to any of the edge weights in this structure will be $0$. However, if we want to change the value of this bit, we will have one of $v_c$ and $v_c'$ output $0$. That will result in a nonzero output from $v_4$ or $v_4'$, which will lead to the net's output having a nonzero derivative with respect to some of the edge weights in this structure. Then, if the net gives the wrong output, the weights of some of the edges in the structure will be multiplied by $-1$, including the weight of the edge from $v_0$ to $v_1$. Unfortunately, if the net gives the right output then the derivative of the loss function with respect to any edge weight will be $0$, which means that any attempt to change a value in memory on that timestep will fail.

More formally, we have the following.

**Lemma 2** (Editing memory when the net gives the wrong output). *Let $0 < s < 1/18\sqrt{3}$, $\gamma = 2^{-244} \cdot 3^{-1643/2} s^{-362}$, and $L(x) = x^2$ for all $x$. Also, let $(f, G)$ be a neural net such that $G$ contains $M_s$ as a subgraph with $v_6$ as $G$'s output vertex, and there are no edges from vertices outside this subgraph to vertices in the subgraph other than $v_0$, $v_c$, and $v_c'$. Now, assume that this neural net is trained using SGD with learning rate $\gamma$ and loss function $L$ for $t$ time steps, and the following hold:*

1. *The sample output is always $\pm 1$.*

2. *The net gives an output of $\pm 1$ in every time step.*

3. *$v_0$ outputs $2$ in every time step.*

4. *$v_c$ and $v_c'$ each output $0$ or $2$ in every time step.*

5. *$v_c'$ outputs $2$ in every time step when the net outputs $1$ and $v_c$ outputs $2$ in every time step when the net outputs $-1$.*

6. *The derivatives of the loss function with respect to the weights of all edges leaving this subgraph are always $0$.*

*Then during the training process, the weight of the edge from $v_0$ to $v_1$ is multiplied by $-1$ during every time step when the net gives the wrong output and $v_c$ and $v_c'$ do not both output $2$, and its weight stays the same during all other time steps.*

*Proof.* More precisely, we claim that the weight of the edge from $v_c$ to $v_4$ and the weight of the edge from $v_c'$ to $v_4'$ never change, and that all of the other edges in $M_s$ only ever change by switching signs. Also, we claim that at the end of any time step, either all of the edges on the path from $v_0$ to $v_2$ have their original weights, or all of them have weights equal to the negatives of their original weights. Furthermore, we claim that the same holds for the edges on each path from $v_2$ to $v_6$.

In order to prove this, we induct on the number of time steps. It obviously holds after $0$ time steps. Now, assume that it holds after $t' - 1$ time steps, and consider time step $t'$. If the net gave the correct output, then the derivative of the loss function with respect to the output is $0$, so none of the weights change.

Now, consider the case where the net outputs $1$ and the correct output is $-1$. By assumption, $v_c'$ outputs $2$ in this time step, so $v_4'$ gets an input of $2^{27} \cdot 3^{91} s^{40}$ from $v_3'$ and an input of $-2^{27} \cdot 3^{91} s^{40}$ from $v_c'$. So, both its output and the derivative of its output with respect to its input are $0$. That means that the same holds for $v_5$, which means that none of the edge weights on this path from $v_2$ to $v_6$ change this time step, and nothing backpropagates through this path. If $v_c$ also outputs $2$, then $v_4$ and $v_5$ output $0$ for the same reason, and none of the edge weights in this copy of $M_s$ change. On the other hand, if $v_c$ outputs $0$, then the output vertex gets an input of $2^{243} \cdot 3^{1641/2} s^{364}$ from $v_5$. The derivative of this input with respect to the weight of the edge from $v_{i-1}$ to $v_i$ is $2^{243} \cdot 3^{1641/2} s^{364} \cdot [3^{6-i}/(3^{3-i/2} s)]$ if these weights are positive, and the negative of that if they are negative. Furthermore, the derivative of the loss function with respect to the input to the output vertex is $12$. So, the algorithm reduces the weights of all the edges on the path from $v_0$ to $v_6$ that goes through $v_4$ exactly enough to change them to the negatives of their former values. Also, since $v_c$ output $0$, the weight of the edge from $v_c$ to $v_4$ had no effect on anything this time step, so it stays unchanged.

The case where the net outputs $-1$ and the correct output is $1$ is analogous, with the modification that the output vertex gets an input of $-2^{243} \cdot 3^{1641/2} s^{364}$ from $v_5'$ if $v_c'$ outputs $0$ and the edges on the path from $v_0$ to $v_6$ that goes through $v_4'$ are the ones that change signs. So, by induction, the claimed properties hold at the end of every time step. Furthermore, this argument shows that the sign of the edge from $v_0$ to $v_1$ changes in exactly the time steps where the net outputs the wrong value and $v_c$ and $v_c'$ do not both output $2$. $\square$

So, $M_s$ satisifes some but not all of the properties we would like a memory component to have. We can read the bit it is storing, and we can control which time steps it might change in by controlling the inputs to $v_c$ and $v_c'$. However, for it to work we need the output of the overall net to be $\pm 1$ in every time step, and each such memory component will input $\pm 2^{243} \cdot 3^{1641/2} s^{364}$ to the output vertex every time we try to flip it. More problematically, the values these components are storing can only

change when the net gets the output wrong. We can deal with the first issue by choosing parameters such that $2^{243} \cdot 3^{1641/2} s^{364}$ is the inverse of an integer that is at least as large as the number of bits that we want to remember, and then adding some extraneous memory components that we can flip in order to ensure that exactly $1/2^{243} \cdot 3^{1641/2} s^{364}$ memory components get flipped in each time step. We cannot change the fact that the net will not learn from samples where it got the output right, but we can use this to emulate any efficient learning algorithm that only updates when it gets something wrong. More formally, we have the following.

**Lemma 3.** *For each $n$, let $m_n$ be polynomial in $n$, and $h_n : \{0,1\}^{n+m_n} \to \{0,1\}$ and $g_n : \{0,1\}^{n+m_n} \to \{0,1\}^{m_n}$ be functions that can be computed in polynomial time. Then there exists a neural net $(G_n, f)$ of polynomial size and $\gamma > 0$ such that the following holds. Let $T > 0$ and $(x_t, y_t) \in \{0,1\}^n \cdot \{0,1\}$ for each $0 < t \leq T$. Then, let $b_0 = (0, ..., 0)$, and for each $0 < t \leq T$, let $y_t^\star = h_n(x_t, b_{t-1})$ and let $b_t$ equal $b_{t-1}$ if $y_t^\star = y_t$ and $g_n(x_t, b_{t-1})$ otherwise. Then if we use stochastic gradient descent to train $(G_n, f)$ on the samples $(2x_t - 1, 2y_t - 1)$ with a learning rate of $\gamma$, the net outputs $1$ in every time step where $y_t^\star = 1$ and $-1$ in every time step where $y_t^\star = 0$.*

*Proof.* First, let $m' = \max(m, \lceil 2^{-243} 3^{-1641/2} (18\sqrt{3})^{364} \rceil)$, and $s = \sqrt[364]{2^{-243} 3^{-1641/2}/m'}$. Then, set $\gamma = 2^{-244} \cdot 3^{-1643/2} s^{-362}$.

We construct $G_n$ as follows. First, we take $m + m'$ copies of $M_s$, merge all of the copies of $v_6$ to make an output vertex, and merge all of the copies of $v_0$. Then we add in $n$ input vertices and a constant vertex and add an edge of weight $2$ from the constant vertex to $v_0$. Next, define $r : \{0,1\}^{n+m} \to \{0,1\}^{1+2m+2m'}$ such that given $x \in \{0,1\}^n$ and $b \in \{0,1\}^m$, $r(x,b)$ lists $h_n(x,b)$ and one half the values of the $v_c$ and $v'_c$ necessary to change the values stored by the first $m$ memory units in the net from $b$ to $g_n(x,b)$ and then flip the next $m' - |\{i : b_i \neq (g_n(x,b))_i\}|$ provided the net outputs $2h_n(x,b) - 1$. Then, add a section to the net that computes $r$ on the input bits and the bits stored in the first $m$ memory units, and connect each copy of $v_c$ or $v'_c$ to the appropriate output by an edge of weight $1/2$ and the constant bit by an edge of weight $1$.

In order to show that this works, first observe that since $h_n$ and $g_n$ can be computed efficiently, so can $r$. So, there exists a polynomial sized subnet that computes it correctly by lemma 1. That lemma also shows that this section of the net will never change as long as all of the inputs and all of the memory bits encode $0$ or $1$ in every time step. Similarly, in every time step $v_0$ will have an input of $2$ and all of the copies of $v_c$ and $v'_c$ will have inputs of $0$ or $2$. So, the derivatives of their outputs with respect to their inputs will be $0$, which means that the weights of the edges leading to them will never change. That means that the only edges that could change in weight are those in the memory components. In each time step, $m'$ memory components each contribute $(2h_n(x_t, b_{t-1}) - 1)/m'$ to the output vertex, so it takes on a value of $(2h_n(x_t, b_{t-1}) - 1)$, assuming that the memory components were storing $b_{t-1}$ like they were supposed to. As such, the net outputs $y_t^\star$, the memory bits stay the same if $y_t^\star = y_t$, and the first $m$ memory bits get changed to $g_n(x_t, b_{t-1})$ otherwise with some irrelevant changes to the rest. Therefore, by induction on the time step, this net performs correctly on all time steps. $\square$

**Remark 9.** *With the construction in this proof, $m'$ will always be at least $10^{79}$, which ensures that this net will be impractically large. This is a result of the fact that the only edges going to the output vertex are those contained in the memory component, and the paths in the memory component take a small activation and repeatedly cube it. If we had chosen an activation function that raises its input to the $\frac{11}{9}$ when its absolute value was less than $1$ instead of cubing it, the minimum possible value of $m'$ would have been on the order of $1000$.*

In other words, we can train a neural net with SGD in order to duplicate any efficient algorithm that takes $n$ bits as input, gives $1$ bit as output, and only updates its memory when its output fails to match some designated "correct" output. The only part of that that is a problem is the restriction that it can not update its memory in steps when it gets the output right. As a result, the probability distribution of the samples that the net actually learns from could be different from the true probability distribution of the samples. We do not know how an algorithm that we are emulating will behave if we draw its samples from a different probability distribution, so this could cause problems. Our solution to that will be to have a training phase where the net gives random outputs so that it will learn from each sample with probability $1/2$, and then switch to attempting to compute the actual correct output rather than learning. That allows us to prove the following (re-statement of Theorem 1).

**Theorem 4.** *For each $n > 0$, let $P_\mathcal{X}$ be a probability measure on $\{0,1\}^n$, and $P_\mathcal{F}$ be a probability measure on the set of functions from $\{0,1\}^n$ to $\{0,1\}$. Also, let $B_{1/2}$ be the uniform distribution on $\{0,1\}$. Next, define $\alpha_n$ such that there is some algorithm that takes a polynomial number of samples $(x_i, F(x_i))$ where the $x_i$ are independently drawn from $P_\mathcal{X}$ and $F \sim P_\mathcal{F}$, runs in polynomial time, and learns $(P_\mathcal{F}, P_\mathcal{X})$ with accuracy $\alpha$. Then there exists $\gamma_n > 0$, a polynomial-sized neural net $(G_n, f)$, and a polynomial $T_n$ such that using stochastic gradient descent with learning rate $\gamma_n$ and loss function $L(x) = x^2$ to train $(G_n, f)$ on $T_n$ samples $((2x_i - 1, 2r_i - 1, 2r_i' - 1), F(x_i))$ where $(x_i, r_i, r_i') \sim P_\mathcal{X} \times B_{1/2}^2$ learns $(P_\mathcal{F}, P_\mathcal{X})$ with accuracy $\alpha - o(1)$.*

*Proof.* We can assume that the algorithm counts the samples it has received, learns from the designated number, and then stops learning if it receives additional samples. The fact that the algorithm learns in polynomial time also means that it can only update a polynomial number of locations in memory, so it only needs a polynomial number of bits of memory, $m_n$. Also, its learning process can be divided into steps which each query at most one new sample $(x_i, F(x_i))$ and one new random bit. So, there must be an efficiently computable function $A$ such that if $b$ is the value of the algorithm's memory at the start of a step, and it receives $(x_i, y_i)$ as its sample (if any) and $r_i$ as its random bit (if any), then it ends the step with its memory set to $A(b, x_i, y_i, r_i)$.

Now, define $A' : \{0,1\}^{m_n + n + 3} \to \{0,1\}^{m_n}$ such that

$$A'(b, x, y, r, r') = \begin{cases} b & \text{if } y = r' \\ A(b, x, y, r) & \text{if } y \neq r' \end{cases}$$

Next, let $b_0$ be the initial state of the algorithm's memory, and consider setting $b_i = A'(b_{i-1}, x_i, F(x_i), r_i, r_i')$ for each $i > 0$. We know that $r_i'$ is equally likely to be $0$ or $1$ and independent of all other components, so $b_i$ is equal to $A(b_{i-1}, x_i, F(x_i), r_i)$ with probability $1/2$ and $b_{i-1}$ otherwise. Furthermore, the probability distribution of $(b_{i-1}, x_i, F(x_i), r_i)$ is independent of whether or not $y_i = r_i'$. Also, if we set $b' = b_0$ and then repeatedly replace $b'$ with $A(b', x, F(x), r)$, then there is some polynomial number of times we need to do that before $b'$ stops changing because the algorithm has enough samples and is no longer learning. So, with probability $1 - o(1)$, the value of $b_i$ will stabilize by the time it has received $n$ times that many samples. Furthermore, the probability distribution of the value $b_i$ stabilizes at is exactly the same as the probability distribution of the value the algorithm's memory stabilizes at because the probability distribution of tuples $(b_{i-1}, x_i, F(x_i))$ that actually result in changes to $b_i$ is exactly the same as the overall probability distribution of $(b_{i-1}, x_i, F(x_i))$. So, given the final value of $b_i$, one can efficiently compute $F$ with an expected accuracy of at least $\alpha$.

Now, let $\overline{A}(b, x)$ be the value the algorithm outputs when trying to compute $F(x)$ if its memory has a value of $b$ after training. Then, define $A''$ such that

$$A''(b, x, r, r') = \begin{cases} \overline{A}(b, x) & \text{if b is a memory state resulting from training on enough samples} \\ r' & \text{otherwise.} \end{cases}$$

By the previous lemma, there exists a polynomial sized neural net $(G_n, f)$ and $\gamma_n > 0$ such that if we use SGD to train $(G_n, f)$ on $((2x_i - 1, 2r_i - 1, 2r_i' - 1), F(x_i))$ with a learning rate of $\gamma_n$ then the net outputs $2A''(b_{i-1}, x_i, r_i, r_i') - 1$ for all $i$. By the previous analysis, that means that after a polynomial number of steps, the net will compute $F$ with an expected accuracy of $\alpha - o(1)$. □

**Remark 10.** *This net uses two random bits because it needs one in order to randomly choose outputs during the learning phase and another to supply randomness in order to emulate randomized algorithms. If we let $m$ be the minimum number of gates in a circuit that computes the algorithm's output and the contents of its memory after the current timestep from its input, its current memory values, and feedback on what the correct output was, then the neural net in question will have $\theta(m)$ vertices and $\gamma_n = \theta(m^{362/364})$. If the algorithm that we are emulating is deterministic, then $T_n$ will be approximately twice the number of samples the algorithm needs to learn the function; if it is randomized it might need a number of additional samples equal to approximately twice the number of random bits the algorithm needs.*

So, for any distribution of functions from $\{0,1\}^n$ to $\{0,1\}$ that can be learned in polynomial time, there is a neural net that learns it in polynomial time when it is trained by SGD.

*Proof of Corollary 1.* Previous theorem shows that each efficiently learnable $(P_\mathcal{F}, P_\mathcal{X})$ has some neural net that learns it efficiently. We will next use a Kolmogorov complexity-like argument to emulate a metaalgorithm as follows:

Learning-Metaalgorithm(c):

1. List every algorithm that can be written in at most $\log(\log(n))$ bits.

2. Get $n^c$ samples from the target distribution, and train each of these algorithms on them in parallel. If any of these algorithms takes more than $n^c$ time steps on any sample, then interrupt it and skip training it on that sample.

3. Get $n^c$ more samples, have all of the aforementioned algorithms attempt to compute the function on each of them, and record which of them was most accurate. Again, if any of them take more than $n^c$ steps on one of these samples, interrupt it and consider it as having computed the function incorrectly on that sample.

4. Return the function that resulted from training the most accurate algorithm.

Given any distribution that is efficiently learnable, there exist $\epsilon, c > 0$ such that there is some algorithm that learns $(P_\mathcal{F}, P_\mathcal{X})$ with accuracy $1/2 + \epsilon - o(1)$, needs at most $n^c$ samples in order to do so, and takes a maximum of $n^c$ time steps on each sample. For all sufficiently large $n$, this algorithm will be less than $\log(\log(n))$ bits long, so Learning-Metaalgorithm(c) will consider it. There are only $O(\log(n))$ algorithms that are at most $\log(\log(n))$ bits long, so in the testing phase all of them will have observed accuracies within $O(n^{-c/2} \log(n))$ of their actual accuracies with high probability. That means that the function that Learning-Metaalgorithm(c) judges as most accurate will be at most $O(n^{-c/2} \log(n))$ less accurate than the true most accurate function considered. So, Learning-Metaalgorithm(c) learns $(P_\mathcal{F}, P_\mathcal{X})$ with accuracy $1/2 + \epsilon - o(1)$. More precisely, this shows that for any efficiently learnable distribution, there exists $C_0$ such that for all $c > C_0$, Learning-Metaalgorithm(c) learns $(P_\mathcal{F}, P_\mathcal{X})$.

Now, if we let $(f, G_c)$ be a neural net emulating Learning-Metaalgorithm(c), then $(f, G_c)$ has polynomial size and can be constructed in polynomial time for any fixed $c$. Any efficiently learnable distribution can be learned by training $(f, G_c)$ with stochastic gradient descent with the right $c$ and the right learning rate, assuming that random bits are appended to the input. Furthermore, the only thing we need to know about the data distribution in order to choose the net and learning rate is some upper bound on the number of samples and amount of time needed to learn it. $\qquad \square$

**Remark 11.** *The previous remark shows that for any $c > 0$, there is a polynomial sized neural net that learns any $(P_\mathcal{F}, P_\mathcal{X})$ that can be learned by an algorithm that uses $n^c$ samples and $n^c$ time per sample. However, that is still more restrictive than we really need to be. It is actually possible to build a net that learns any $(P_\mathcal{F}, P_\mathcal{X})$ that can be efficiently learned using $n^c$ memory, and then computed in $n^c$ time once the learning process is done. In order to show this, first observe that any learning algorithm that spends more than $n^c$ time on each sample can be rewritten to simply get a new sample and ignore it after every $n^c$ steps. That converts it to an algorithm that spends $n^c$ time after receiving each sample while multiplying the number of samples it needs by an amount that is at most polynomial in $n$.*

*The fact that we do not know how many samples the algorithm needs can be dealt with by modifying the metaalgorithm to find the algorithm that performs best when trained on 1 sample, then the algorithm that performs best when trained on 2, then the algorithm that performs best when trained on 4, and so on. That way, after receiving any number of samples, it will have learned to compute the function with an accuracy that is within $o(1)$ of the best accuracy attainable after learning from $1/4$ that number of samples. The fact that we do not know how many samples we need also renders us unable to have a learning phase, and then switch to attempting to compute the function accurately after we have seen enough samples. Instead, we need to have it try to learn from each sample with a gradually decreasing probability and try to compute the function otherwise. For instance, consider designing the net so that it keeps a count of exactly how many times it has been wrong. Whenever that number reaches a perfect square, it attempts to learn from the next sample; otherwise, it tries to compute the function on that input. If it takes the metaalgorithm $n^{c'}$ samples to learn the function with accuracy $1 - \epsilon$, then it will take this net roughly $n^{2c'}$ samples to learn it with the same accuracy,*

*and by that point the steps where it attempts to learn the function rather than computing it will only add another $o(1)$ to the error rate. So, if there is any efficient algorithm that learns $(P_\mathcal{F}, P_\mathcal{X})$ with $n^c$ memory and computes it in $n^c$ time once it has learned it, then this net will learn it efficiently.*

*Finally, it is necessary to know $c$ in order to obtain such a universality result, since given a neural net of size $O(n^{c'})$, one could simply pick a function that requires a net of size $\Omega(n^{c'+1})$ to compute.*

## 4.2 Noisy emulation of arbitrary algorithms

So far, our discussion of emulating arbitrary learning algorithms using SGD has assumed that we are using SGD without noise. It is of particular interest to ask whether there are efficiently learnable functions that noisy SGD can never learn with inverse-polynomial noise, as perfect GD or SQ algorithms break in such cases (e.g., parities). It turns out that the emulation argument can be adapted to sufficiently small amounts of noise.

The computation component is already fairly noise tolerant because the inputs to all of its vertices will normally always have absolute values of at least $2$. If these are changed by less than $1/2$, these vertices will still have activations of $\pm 2$ with the same signs as before, and the derivatives of their activations with respect to their inputs will remain $0$.

However, the memory component has more problems handling noise. In the noise-free case, whenever we do not want the value it stores to change, we arrange for some key vertices inside the component to receive input $0$ so that their outputs and the derivatives of their outputs with respect to their inputs will both be $0$. However, once we start adding noise we will no longer be able to ensure that the inputs to these vertices are exactly $0$. This could result in a feedback loop where the edge weights shift faster and faster as they get further away from their desired values. In order to avoid this, we will use an activation function designed to have output $0$ whenever its input is sufficiently close to $0$. More precisely, in this section we will use an activation function $f^\star : \mathbb{R} \to \mathbb{R}$ chosen so that $f^\star(x) = 0$ whenever $|x| \le 2^{-121}3^{-9}$, $f^\star(x) = x^3$ whenever $2^{-120}3^{-9} \le |x| \le 1$, and $f^\star(x) = 2 \operatorname{sign}(x)$ whenever $|x| \ge 3/2$. There must be a way to define $f^\star$ on the remaining intervals such that it is smooth and nondecreasing. The details of how this is done will not affect out argument, so we pick some such assignment.

The memory component also has trouble handling bit flips when there is noise. Any time we flip a bit stored in memory, any errors in the edge weights of the copy of $M_s$ storing that bit are likely to get worse. As a result, making the memory component noise tolerant requires a fairly substantial redesign. First of all, in order to prevent perturbations in its edge weights from being amplified until they become major problems, we will only update each value stored in memory once. That still leaves the issue that due to errors in the edge weights, we cannot ensure that the output of the net is exactly $\pm 1$. As a result, even if the net gets the output right, the edge weights will still change somewhat. That introduces the possibility that multiple unsuccessful attempts at flipping a bit in memory will eventually cause major distortions to the corresponding edge weights. In order to address that, we will have our net always give an output of $1/2$ during the learning phase so that whenever we try to change a value in memory, it will change significantly regardless of what the correct output is. Of course, that leaves each memory component with $3$ possible states, the state it is in originally, the state it changes to if the correct output is $1$, and the state it changes to if the correct output is $-1$. More precisely, each memory value will be stored in a copy of the following.

**Definition 7.** *Let $M'$ be the weighted directed graph with $9$ vertices, $v_0$, $v_1$, $v_2$, $v_3$, $v_4$, $v_5$, $v_c$, $v'_c$, and $v_r$ and the following edges:*

1. *An edge of weight $3^{-t/2}/4$ from $v_t$ to $v_{t+1}$ for each $t$*

2. *An edge of weight $128$ from $v_1$ to $v_r$*

3. *An edge of weight $-2^{-81} \cdot 3^{-9}$ from $v_c$ to $v_4$*

4. *An edge of weight $-2^{-41} \cdot 3^{-9}$ from $v'_c$ to $v_4$*

See Figure 3 for a representation of $M'$. The idea is that by controlling the values of $v_c$ and $v'_c$ we can either force $v_4$ to have an input of approximately $0$ in order to prevent any of the weights from changing or allow it to have a significant value in which case the weights will change. With the correct learning rate, if the correct output is $1$ then the weights of the edges on the path from $v_0$ to $v_5$

Figure 3: The noise-tolerant memory component $M'$.

will double, while if the correct output is $-1$ then these weights will multiply by $-2$. That means that $v_2$ will have an output of approximately $2^{-24}3^{-3/2}$ if this has never been changed, and an output of approximately $2^{-12}3^{-3/2}$ if it has. Meanwhile, $v_r$ will have an output of $-2$ if it was changed when the correct output was $-1$ and a value of $2$ otherwise. More formally, we have the following.

**Lemma 4** (Editing memory using noisy SGD). *Let $\gamma = 2^{716/3} \cdot 3^{24}$, and $L(x) = x^2$ for all $x$. Next, let $t_0, T \in \mathbb{Z}^+$ and $0 < \epsilon, \epsilon'$ such that $\epsilon \leq 2^{-134}3^{-11}$, $\epsilon' \leq 2^{-123}3^{-11}$. Also, let $(f^\star, G)$ be a neural net such that $G$ contains $M'$ as a subgraph with $v_5$ as $G$'s output vertex, $v_0$ as the constant vertex, and no edges from vertices outside this subgraph to vertices in the subgraph other than $v_c$ and $v'_c$. Now, assume that this neural net is trained using noisy SGD with learning rate $\gamma$ and loss function $L$ for $T - 1$ time steps, and then evaluated on an input, and the following hold:*

1. *The sample label is always $\pm 1$.*

2. *The net gives an output that is in $[1/2 - \epsilon', 1/2 + \epsilon']$ on step $t$ for every $t < T$.*

3. *For every $t < t_0$, $v_c$ gives an output of $2$ and $v'_c$ gives an output of $0$ on step $t$.*

4. *If $t_0 \leq T$ then $v_c$ and $v'_c$ both give outputs of $0$ on step $t_0$.*

5. *For every $t > t_0$, $v'_c$ gives an output of $2$ and $v_c$ gives an output of $0$ on step $t$.*

6. *For each edge in the graph, the sum of the absolute values of the noise terms applied to that edge over the course of the training process is at most $\epsilon$.*

7. *The derivatives of the loss function with respect to the weights of all edges leaving this subgraph are always $0$.*

*Then during the training process, $v_2$ gives an output in $[2^{-25}3^{-3/2}, 2^{-23}3^{-3/2}]$ on step $t$ for all $t \leq t_0$ and an output in $[2^{-13}3^{-3/2}, 2^{-11}3^{-3/2}]$ on step $t$ for all $t > t_0$. Also, on step $t$, $v_r$ gives an output of $-2$ if $t > t_0$ and the sample label was $-1$ on step $t_0$ and an output of $2$ otherwise. Thirdly, on step $t_0$ the edge from $v_4$ to $v_5$ provides an input to the output vertex in $[2^{-242}3^{-29} - 2^{-201}3^{-27}\epsilon, 2^{-242}3^{-29} + 2^{-201}3^{-27}\epsilon]$, and for all $t \neq t_0$, the edge from $v_4$ to $v_5$ provides an input of $0$ to the output vertex on step $t$.*

*Proof.* First of all, we define the target weight of an edge to be what we would like its weight to be. More precisely, the target weights of $(v_c, v_4)$, $(v'_c, v_4)$, and $(v_1, v_r)$ are defined to be equal to their initial weights at all time steps. The target weights of the edges on the path from $v_0$ to $v_5$ are defined to be equal to their initial weights until step $t_0$. After step $t_0$, these edges have target weights that are equal to double their initial weights if the sample label at step $t_0$ was $1$ and $-2$ times their initial weights if the sample label at step $t_0$ was $-1$.

Next, we define the primary distortion of a given edge at a given time to be the sum of all noise terms added to its weight by noisy SGD up to that point. Then, we define the secondary distortion of an edge to be the difference between its weight, and the sum of its target weight and its primary

distortion. By our assumptions, the primary distortion of any edge always has an absolute value of at most $\epsilon$. We plan to prove that the secondary distortion stays reasonably small by inducting on the time step, at which point we will have established that the actual weights of the edges stay reasonably close to their target weights.

Now, for all vertices $v$ and $v'$, and every time step $t$, let $w_{(v,v')[t]}$ be the weight of the edge from $v$ to $v'$ at the start of step $t$, $y_{v[t]}$ be the output of $v$ on step $t$, $d_{v[t]}$ be the derivative of the loss function with respect to the output of $v$ on step $t$, and $d'_{v[t]}$ be the derivative of the loss function with respect to the input of $v$ on step $t$. Next, consider some $t < t_0$ and assume that the secondary distortion of every edge in $M'$ is $0$ at the start of step $t$. In this case, $v_1$ has an activation in $[(1/4 - \epsilon)^3, (1/4 + \epsilon)^3]$, so $v_r$ has an activation of $2$ and the derivative of the loss function with respect to $w_{(v_1,v_r)}$ is $0$. Also, the activation of $v_2$ is between $2^{-25}3^{-3/2}$ and $2^{-23}3^{-3/2}$. On another note, the total input to $v_4$ on step $t$ is

$$w_{(v_0,v_1)[t]}^{27} w_{(v_1,v_2)[t]}^{9} w_{(v_2,v_3)[t]}^{3} w_{(v_3,v_4)[t]} + w_{(v_c,v_4)[t]} y_{v_c[t]} + w_{(v'_c,v_4)[t]} y_{v'_c[t]}$$

$$\leq \left(\frac{1}{4} + \epsilon\right)^{27} \left(\frac{\sqrt{3}}{12} + \epsilon\right)^{9} \left(\frac{1}{12} + \epsilon\right)^{3} \left(\frac{\sqrt{3}}{36} + \epsilon\right) + \left(-2^{-81}3^{-9} + \epsilon\right) \cdot 2$$

$$\leq 2^{-80}3^{-9}e^{(144 + 48\sqrt{3})\epsilon} - 2^{-80}3^{-9} + 2\epsilon$$

$$\leq 3\epsilon$$

On the flip side, the total input to $v_4$ on step $t$ is at least

$$\left(\frac{1}{4} - \epsilon\right)^{27} \left(\frac{\sqrt{3}}{12} - \epsilon\right)^{9} \left(\frac{1}{12} - \epsilon\right)^{3} \left(\frac{\sqrt{3}}{36} - \epsilon\right) + \left(-2^{-81}3^{-9} - \epsilon\right) \cdot 2$$

$$\geq 2^{-80}3^{-9}e^{-2(144 + 48\sqrt{3})\epsilon} - 2^{-80}3^{-9} - 2\epsilon$$

$$\geq -3\epsilon$$

So, $|y_{v_4[t]}| = 0$, and the edge from $v_4$ to $v_5$ provides an input of $0$ to the output vertex on step $t$. The derivative of this contribution with respect to the weights of any of the edges in $M'$ is also $0$. So, if all of the secondary distortions are $0$ at the beginning of step $t$, then all of the secondary distortions will still be $0$ at the end of step $t$. The secondary distortions start at $0$, so by induction on $t$, the secondary distortions are all $0$ at the end of step $t$ for every $t < \min(t_0, T)$. This also implies that the edge from $v_4$ to $v_5$ provides an input of $0$ to the output, $y_{v_r[t]} = 2$, and $v_{2[t]} \in [2^{-25}3^{-3/2}, 2^{-23}3^{-3/2}]$ for every $t < t_0$.

Now, consider the case where $t = t_0 \leq T$. In this case, $v_r$ has an activation of $2$ and the derivative of the loss function with respect to $w_{(v_1,v_r)}$ is $0$ for the same reasons as in the last case. Also, the activation of $v_2$ is still between $2^{-25}3^{-3/2}$ and $2^{-23}3^{-3/2}$.

On this step, the total input to $v_4$ is

$$w_{(v_0,v_1)[t]}^{27} w_{(v_1,v_2)[t]}^{9} w_{(v_2,v_3)[t]}^{3} w_{(v_3,v_4)[t]} + w_{(v_c,v_4)[t]} y_{v_c[t]} + w_{(v'_c,v_4)[t]} y_{v'_c[t]}$$

$$\leq \left(\frac{1}{4} + \epsilon\right)^{27} \left(\frac{\sqrt{3}}{12} + \epsilon\right)^{9} \left(\frac{1}{12} + \epsilon\right)^{3} \left(\frac{\sqrt{3}}{36} + \epsilon\right) + 0$$

$$\leq 2^{-80}3^{-9}e^{(144 + 48\sqrt{3})\epsilon}$$

$$\leq 2^{-80}3^{-9} + 2^{-74}3^{-7}\epsilon$$

On the flip side, the total input to $v_4$ is at least

$$\left(\frac{1}{4} - \epsilon\right)^{27} \left(\frac{\sqrt{3}}{12} - \epsilon\right)^{9} \left(\frac{1}{12} - \epsilon\right)^{3} \left(\frac{\sqrt{3}}{36} - \epsilon\right) + 0$$

$$\geq 2^{-80}3^{-9}e^{-2(144 + 48\sqrt{3})\epsilon}$$

$$\geq 2^{-80}3^{-9} - 2^{-74}3^{-7}\epsilon$$

So, $y_{v_4[t]} \in [(2^{-80}3^{-9} - 2^{-74}3^{-7}\epsilon)^3, (2^{-80}3^{-9} + 2^{-74}3^{-7}\epsilon)^3]$, and the edge from $v_4$ to $v_5$ provides an input in $[2^{-242}3^{-29} - 2^{-235}3^{-26}\epsilon, 2^{-242}3^{-29} + 2^{-235}3^{-26}\epsilon]$ to the output vertex on step $t_0$. If $t_0 < T$ then the net gives an output in $[1/2 - \epsilon', 1/2 + \epsilon']$, so $d_{v_5[t]}$ is in $[-1 - 2\epsilon', -1 + 2\epsilon']$ if the sample label is $1$ and in $[3 - 2\epsilon', 3 + 2\epsilon']$ if the sample label is $-1$. That in turn means that $d'_{v_5[t]}$ is in $[\frac{3\sqrt[3]{2}}{2}(-1 - 4\epsilon'), \frac{3\sqrt[3]{2}}{2}(-1 + 4\epsilon')]$ if the sample label is $1$ and in $[\frac{3\sqrt[3]{2}}{2}(3 - 8\epsilon'), \frac{3\sqrt[3]{2}}{2}(3 + 8\epsilon')]$ if the sample label is $-1$. Either way, the derivatives of the loss function with respect to $w_{(v_c,v_4)}$ and $w_{(v'_c,v_4)}$ are both $0$.

Also, for each $0 \le i < 5$, the derivative of the loss function with respect to $w_{(v_i,v_{i+1})}$ is

$$w^{81}_{(v_0,v_1)[t]}w^{27}_{(v_1,v_2)[t]}w^9_{(v_2,v_3)[t]}w^3_{(v_3,v_4)[t]}w_{(v_4,v_5)} \cdot \frac{3^{4-i}}{w_{(v_i,v_{i+1})}} \cdot d'_{v[5]}$$

which is between $2^{-240}3^{-29} \cdot (1 - 7200\epsilon) \cdot 3^{4-i/2} \cdot d'_{v[5]}$ and $2^{-240}3^{-29} \cdot (1 + 7200\epsilon) \cdot 3^{4-i/2} \cdot d'_{v[5]}$

So, if the sample label is $1$, then on this step gradient descent increases the weight of each edge on the path from $v_0$ to $v_5$ by an amount that is within $3600\epsilon + 2\epsilon'$ of its original value. If the sample label is $-1$, then on this step gradient descent decreases the weight of each edge on this path by an amount that is within $10800\epsilon + 6\epsilon'$ of thrice its original value. Either way, it leaves the weight of the edge from $v_1$ to $v_r$ unchanged. So, all of the secondary distortions will be at most $10800\epsilon + 6\epsilon'$ at the end of step $t_0$ if $t_0 < T$.

Finally, consider the case where $t > t_0$ and assume that the secondary distortion of every edge in $M'$ is at most $10800\epsilon + 6\epsilon'$ at the start of step $t$. Also, let $\epsilon'' = 10801\epsilon + 6\epsilon'$, and $y_0$ be the sample label from step $t_0$. In this case, $v_1$ has an activation between $(1/2 - \epsilon'')^3 y_0$ and $(1/2 + \epsilon'')^3 y_0$, so $v_r$ has an activation of $2y_0$ and the derivative of the loss function with respect to $w_{(v_1,v_r)}$ is $0$. Also, the activation of $v_2$ is between $2^{-13}3^{-3/2}$ and $2^{-11}3^{-3/2}$. On another note, the total input to $v_4$ on step $t$ is

$$w^{27}_{(v_0,v_1)[t]}w^9_{(v_1,v_2)[t]}w^3_{(v_2,v_3)[t]}w_{(v_3,v_4)[t]} + w_{(v_c,v_4)[t]}y_{v_c[t]} + w_{(v'_c,v_4)[t]}y_{v'_c[t]}$$

$$\le \left(\frac{y_0}{2} + \epsilon''y_0\right)^{27}\left(\frac{\sqrt{3}y_0}{6} + \epsilon''y_0\right)^9\left(\frac{y_0}{6} + \epsilon''y_0\right)^3\left(\frac{\sqrt{3}y_0}{18} + \epsilon''y_0\right) + (-2^{-41}3^{-9} + \epsilon'') \cdot 2$$

$$\le 2^{-40}3^{-9}e^{(72+24\sqrt{3})\epsilon''} - 2^{-40}3^{-9} + 2\epsilon''$$

$$\le 3\epsilon''$$

On the flip side, the total input to $v_4$ on step $t$ is at least

$$\left(\frac{y_0}{2} - \epsilon''y_0\right)^{27}\left(\frac{\sqrt{3}y_0}{6} - \epsilon''y_0\right)^9\left(\frac{y_0}{6} - \epsilon''y_0\right)^3\left(\frac{\sqrt{3}y_0}{18} - \epsilon''y_0\right) + (-2^{-41}3^{-9} - \epsilon'') \cdot 2$$

$$\ge 2^{-40}3^{-9}e^{-2(72+24\sqrt{3})\epsilon''} - 2^{-40}3^{-9} - 2\epsilon''$$

$$\ge -3\epsilon''$$

So, $y_{v_4[t]} = 0$, and the edge from $v_4$ to $v_5$ provides an input of $0$ to the output vertex on step $t$. The derivatives of this contribution with respect to the weights of any of the edges in $M'$ are also $0$. So, if all of the secondary distortions are at most $10800\epsilon + 6\epsilon'$ at the beginning of step $t$, then all of the secondary distortions will still be at most $10800\epsilon + 6\epsilon'$ at the end of step $t$. We have already established that the secondary distortions will be in that range at the end of step $t_0$, so by induction on $t$, the secondary distortions are all at most $10800\epsilon + 6\epsilon'$ at the end of step $t$ for every $t_0 < t < T'$. This also implies that the edge from $v_4$ to $v_5$ provides an input of $0$ to the output, $y_{v_r[t]} = 2y_0$ and $v_{v_2[t]} \in [2^{-13}3^{-3/2}, 2^{-11}3^{-3/2}]$ for every $t > t_0$.

$\square$

Now that we have established that we can use $M'$ to store information in a noise tolerant manner, our next order of business is to show that we can make the computation component noise-tolerant.

This is relatively simple because all of its vertices always have inputs of absolute value at least 2, so changing these inputs by less than $1/2$ has no effect. We have the following.

**Lemma 5** (Backpropagation-proofed noise-tolerant circuit emulation)**.** *Let $h : \{0,1\}^m \to \{0,1\}^{m'}$ be a function that can be computed by a circuit made of AND, OR, and NOT gates with a total of $b$ gates. Also, consider a neural net with $m$ input[16] vertices $v'_1, ..., v'_m$, and choose real numbers $y^{(0)} < y^{(1)}$. It is possible to add a set of at most $b$ new vertices to the net, including output vertices $v''_1, ..., v''_{m'}$, along with edges leading to them such that for any possible addition of edges leading from the new vertices to old vertices, if the net is trained by noisy SGD, the output of $v'_i$ is either less than $y^{(0)}$ or more than $y^{(1)}$ for every $i$ in every timestep, and for every edge leading to one of the new vertices, the sum of the absolute values of the noise terms applied to that edge over the course of the training process is less than $1/12$, then the following hold:*

1. *The derivative of the loss function with respect to the weight of each edge leading to a new vertex is $0$ in every timestep, and no paths through the new vertices contribute to the derivative of the loss function with respect to edges leading to the $v'_i$.*

2. *In any given time step, if the output of $v'_i$ encodes $x_i$ with values less than $y^{(0)}$ and values greater than $y^{(1)}$ representing $0$ and $1$ respectively for each $i$[17], then the output of $v''_j$ encodes $h_j(x_1, ..., x_m)$ for each $j$ with $-2$ and $2$ encoding $0$ and $1$ respectively.*

*Proof.* In order to do this, we will add one new vertex for each gate and each input in a circuit that computes $h$. When the new vertices are used to compute $h$, we want each vertex to output 2 if the corresponding gate or input outputs a $1$ and $-2$ if the corresponding gate or input outputs a $0$. In order to do that, we need the vertex to receive an input of at least $3/2$ if the corresponding gate outputs a $1$ and an input of at most $-3/2$ if the corresponding gate outputs a $0$. No vertex can ever give an output with an absolute value greater than 2, and by assumption none of the edges leading to the new vertices will have their weights changed by $1/12$ or more by the noise. As such, any noise terms added to the weights of edges leading to a new vertex will alter its input by at most $1/6$ of its in-degree. So, as long as its input without these noise terms has the desired sign and an absolute value of at least $3/2$ plus $1/6$ of its in-degree, it will give the desired output.

In order to make one new vertex compute the NOT of another new vertex, it suffices to have an edge of weight $-1$ to the vertex computing the NOT and no other edges to that vertex. We can compute an AND of two new vertices by having a vertex with two edges of weight 1 from these vertices and an edge of weight $-2$ from the constant vertex. Similarly, we can compute an OR of two new vertices by having a vertex with two edges of weight 1 from these vertices and an edge of weight 2 from the constant vertex. For each $i$, in order to make a new vertex corresponding to the $i$th input, we add a vertex and give it an edge of weight $4/(y^{(1)} - y^{(0)})$ from the associated $v'_i$ and an edge of weight $-(2y^{(1)} + 2y^{(0)})/(y^{(1)} - y^{(0)})$ from the constant vertex. These provide an overall input of at least 2 to the new vertex if $v'_i$ has an output greater than $y^{(1)}$ and an input of at most $-2$ if $v'_i$ has an output less than $y^{(0)}$.

This ensures that if the outputs of the $v'_i$ encode binary values $x_1, ..., x_m$ appropriately, then each of the new vertices will output the value corresponding to the output of the appropriate gate or input. So, these vertices compute $h(x_1, ..., x_m)$ correctly. Furthermore, since the input to each of these vertices is outside of $(-3/2, 3/2)$, the derivatives of their activation functions with respect to their inputs are all 0. As such, the derivative of the loss function with respect to any of the edges leading to them is always 0, and paths through them do not contribute to changes in the weights of edges leading to the $v'_i$. □

Now that we know that we can make the memory component and computation component work, it is time to put the pieces together. We plan to have the net simply memorize each sample it receives until it has enough information to compute the function. More precisely, if there is an algorithm that needs $T$ samples to learn functions from a given distribution, our net will have $2nT$ copies of $M'$

corresponding to every combination of a timestep $1 \leq t \leq T$, an input bit, and a value for said bit. Then, in step $t$ it will set the copies of $M'$ corresponding to the inputs it received in that time step. That will allow the computation component to determine what the current time step is, and what the inputs and labels were in all previous times steps by checking the values of the copies of $v_2$ and $v_r$. That will allow it to either determine which copies of $M'$ to set next, or attempt to compute the function on the current input and return it. This design works in the following sense.

**Lemma 6.** *For each $n > 0$, let $t_n$ be a positive integer such that $t_n = \omega(1)$ and $t_n = O(n^c)$ for some constant c. Also, let $h_n : \{0,1\}^{(n+1)t_n+n} \to \{0,1\}$ be a function that can be computed in time polynomial in n. Then there exists a polynomial sized neural net $(G_n, f)$ such that the following holds. Let $\gamma = 2^{716/3} \cdot 3^{24}$, $\delta \in [-1/(n^2 t_n), 1/(n^2 t_n)]^{t_n \times |E(G_n)|}$ (i.e., each weight at each time step has a precision error of $1/(n^2 t_n)$), $x^{(i)} \in \{0,1\}^n$ for all $0 \leq i \leq t_n$, and $y^{(i)} \in \{0,1\}$ for all $0 \leq i < t_n$. Then if we use perturbed stochastic gradient descent with noise $\delta$, loss function $L(x) = x^2$, and learning rate $\gamma$ to train $(G_n, f)$ on $(2x^{(i)} - 1, 2y^{(i)} - 1)$ for $0 \leq i < t_n$ and then run the resulting net on $2x_{t,n} - 1$, we will get an output within $1/2$ of $2h\left(x^{(0)}, y^{(0)}, x^{(1)}, y^{(1)}, ..., x^{(t_n)}\right) - 1$ with probability $1 - o(1)$.*

*Proof.* We construct $G_n$ as follows. We start with a graph consisting of $n$ input vertices. Then, we take $2nt_n$ copies of $M'$, merge all of the copies of $v_0$ to make a constant vertex, and merge all of the copies of $v_5$ to make an output vertex. We assign each of these copies a distinct label of the form $M'_{t',i,z}$, where $0 \leq t' < t_n$, $0 < i \leq n$, and $z \in \{0,1\}$. We also add edges of weight 1 from the constant vertex to all of the control vertices. Next, for each $0 \leq t' < t_n$, we add an output control vertex $v_{oc[t']}$. For each such $t'$, we add an edge of weight 1 from the constant vertex to $v_{oc[t']}$ and an edge of weight $\sqrt[3]{4}/4 - 2^{-243}3^{-29}n$ from $v_{oc[t']}$ to the output vertex. Then, we add a final output control vertex $v_{oc[t_n]}$. We do not add an edge from the constant vertex to $v_{oc[t_n]}$, and the edge from $v_{oc[t_n]}$ to the output vertex has weight $49/100$.

Finally, we use the construction from the previous lemma to build a computation component. This component will get input from all of the input vertices and every copy of $v_r$ and $v_2$ in any of the copies of $T'$, interpreting anything less than $2^{-21}3^{-3/2}$ as a 0 and anything more than $2^{-15}3^{-3/2}$ as a 1. This should allow it to read the input bits, and determine which of the copies of $M'$ have been set and what the sample outputs were when they were set. For each control vertex from a copy of $T'$ and each of the first $n$ output control vertices, the computation component will contain a vertex with an edge of weight $1/2$ leading to that vertex. It will contain two vertices with edges of weight $1/2$ leading to $v_{oc[t_n]}$. This should allow it to set each control vertex or output control vertex to 0 or 2, and to set $v_{oc[t_n]}$ to $-2$, 0, or 2.

The computation component will be designed so that in each time step it will do the following, assuming that its edge weights have not changed too much and the outputs of the copies of $v_r$ and $v_2$ are in the ranges given by lemma 4. First, it will determine the smallest $0 \leq t \leq t_n$ such that $M'_{(t',i,z)}$ has not been set for any $t' \geq t$, $0 < i \leq n$, and $z \in \{0,1\}$. That should equal the current timestep. If $t < t_n$, then it will do the following. For each $0 < i \leq n$, it will use the control vertices to set $M'_{(t,i,[x'_i+1]/2)}$, where $x'_i$ is the value it read from the $i$th input vertex. It will keep the rest of the copies of $M'$ the same. It will also attempt to make $v_{oc[t]}$ output 2 and the other output control vertices output 0. If $t = t_n$, then for each $0 \leq t' < t$ and $1 \leq i \leq n$, the computation component will set $x_i^{\star(t')}$ to 1 if $M'_{(t',i,1)}$ has been set, and 0 otherwise. It will set $y^{\star(t')}$ to 1 if either $M'_{(t',1,0)}$ or $M'_{(t',1,1)}$ has been set in a timestep when the sample label was 1 and 0 otherwise. It will also let $x^{\star(t_n)}$ be the values of $x^{(t_n)}$ inferred from the input. Then it will attempt to make $v_{oc[t_n]}$ output $4h(x^{\star(0)}, y^{\star(0)}, ..., x^{\star(t_n)}) - 2$ and the other output control vertices output 0. It will not set any of the copies of $M'$ in this case.

In order to prove that this works, we start by setting $\epsilon = \min(2^{-134}3^{-11}, 2^{77}3^{15}/n)$ and $\epsilon' = 2^{-123}3^{-11}$. The absolute value of the noise term applied to every edge in every time step is at most $1/n^2 t_n$, so the sums of the absolute values of the noise terms applied to every edge over the course of the algorithm are at most $\epsilon$ if $n > 2^{67}3^6$. For the rest of the proof, assume that this holds.

Now, we claim that for every $0 \leq t' < t_n$, all of the following hold:

1. Every copy of $v_r$ or $v_2$ in the memory component outputs a value that is not in $[2^{-21}3^{-3/2}, 2^{-15}3^{-3/2}]$ on timestep $t'$.

2. For every copy of $M'$, there exists $t_0$ such that its copies of $v_c$ and $v'_c$ take on values satisfying lemma 4 for timesteps 0 through $t'$.

3. The net gives an output in $[1/2 - \epsilon', 1/2 + \epsilon']$ on timestep $t'$.

4. The weight of every edge leading to an output control vertex ends step $t'$ with a weight that is within $\epsilon$ of its original weight.

5. For every $t'' > t'$, the weight of the edge from $v_{oc[t'']}$ to the output vertex has a weight within $\epsilon$ of its original weight at the end of step $t'$.

In order to prove this, we use strong induction on $t'$. So, let $0 \le t' < t_n$, and assume that this holds for all $t'' < t'$. By assumption, the conditions of lemma 4 were satisfied for every copy of $M'$ in the first $t'$ timesteps. So, the outputs of the copies of $v_r$ and $v_2$ encode information about their copies of $M'$ in the manner given by this lemma. In particular, that means that their outputs are not in $[2^{-21}3^{-3/2}, 2^{-15}3^{-3/2}]$ on timestep $t'$. By the previous lemma, the fact that this holds for timesteps 0 through $t'$ means that the computation component will still be working properly on step $t'$, it will be able to interpret the inputs it receives correctly, and its output vertices will take on the desired values. The assumptions also imply that every copy of $v_c$ or $v'_c$ took on values of 0 or 2 in step $t''$ for every $t'' < t'$. That means that the derivatives of the loss function with respect to the weights of the edges leading to these vertices was always 0, so their weights at the start of step $t'$ were within $\epsilon$ of their initial weights. That means that the inputs to these copies will be in $[-4\epsilon, 4\epsilon]$ for ones that are supposed to output 1 and in $[2 - 4\epsilon, 2 + 4\epsilon]$ for ones that are supposed to output 2. Between this and the fact that the computation component is working correctly, we have that for each $(t'', i, z)$, the copies of $v_c$ and $v'_c$ in $M'_{(t'', i, z)}$ will have taken on values satisfying the conditions of lemma 4 in timesteps 0 through $t'$ with $t_0$ set to $t''$ if $x_i^{(t'')} = z$ and $t_n + 1$ otherwise.

Similarly, the fact that the weights of the edges leading to the output control vertices stay within $\epsilon$ of their original values for the first $t' - 1$ steps implies that $v_{oc[t'']}$ outputs 2 and all other output control vertices output 0 on step $t''$ for all $t'' \le t'$. That in turn implies that the derivatives of the loss function with respect to these weights were 0 for the first $t' + 1$ steps, and thus that their weights are still within $\epsilon$ of their original values at the end of step $t'$. Now, observe that there are exactly $n$ copies of $M'$ that get set in step $t'$, and each of them provide an input to the output vertex in $[2^{-242}3^{-29} - 2^{-201}3^{-27}\epsilon, 2^{-242}3^{-29} + 2^{-201}3^{-27}\epsilon]$. Also, $v_{oc[t']}$ provides an input to the output in $[\sqrt[3]{4}/2 - 2^{-242}3^{-29}n - 2\epsilon, \sqrt[3]{4}/2 - 2^{-242}3^{-29}n + 2\epsilon]$ on step $t'$, and all other vertices with edges to the output vertex output 0 in this time step. So, the total input to the output vertex is within $2^{-201}3^{-27}\epsilon n + 2\epsilon \le \epsilon'/3$ of $\sqrt[3]{4}/2$. So, the net gives an output in $[1/2 - \epsilon', 1/2 + \epsilon']$ on step $t'$, as desired. This also implies that the derivative of the loss function with respect to the weights of the edges from all output vertices except $v_{oc[t']}$ to the output vertex are 0 on step $t'$. So, for every $t'' > t'$, the weight of the edge from $v_{oc[t'']}$ to the output vertex is still within $\epsilon$ of its original value at the end of step $t'$. This completes the induction argument.

This means that on step $t_n$, all of the copies of $v_r$ and $v_c$ will still have outputs that encode whether or not they have been set and what the sample output was on the steps when they were set in the manner specified in lemma 4, and that the computation component will still be working. So, the computation component will set $x^{\star(t')} = x^{(t')}$ and $y^{\star(t')} = y^{(t')}$ for each $t' < t_n$. It will also set $x^{\star(t_n)} = x^{(t_n)}$, and then it will compute $h\left(x^{(0)}, y^{(0)}, x^{(1)}, y^{(1)}, ..., x^{(t_n)}\right)$ correctly. Call this expression $y'$. All edges leading to the output control and control vertices will still have weights within $\epsilon$ of their original values, so it will be able to make $v_{oc[t_n]}$ output $4y' - 2$, all other output control vertices output 0, and none of the copies of $M'$ provide a nonzero input to the output vertex. The output of $v_{oc[t_n]}$ is 0 in all timesteps prior to $t_n$, so the weight of the edge leading from it to the output vertex at the start of step $t_n$ is within $\epsilon$ of its original value. So, the output vertex will receive a total input that is within $2\epsilon$ of $\frac{49}{50}(2y' - 1)$, and give an output that is within $6\epsilon$ of $\frac{49^3}{50^3}(2y' - 1)$. That is within $1/2$ of $2y' - 1$, as desired. $\qed$

This allows us to prove that we can emulate an arbitrary algorithm by using the fact that the output of any efficient algorithm can be expressed as an efficiently computable function of its inputs and some random bits. More formally, we have the following (re-statement of Theorem 2).

**Theorem 5.** *For each $n > 0$, let $P_\mathcal{X}$ be a probability measure on $\{0,1\}^n$, and $P_\mathcal{F}$ be a probability measure on the set of functions from $\{0,1\}^n$ to $\{0,1\}$. Also, let $B_{1/2}$ be the uniform distribution on $\{0,1\}$, $t_n$ be polynomial in $n$, and $\delta \in [-1/n^2 t_n, 1/n^2 t_n]^{t_n \times |E(G_n)|}$, $x^{(i)} \in \{0,1\}^n$. Next, define $\alpha_n$ such that there is some algorithm that takes $t_n$ samples $(x_i, F(x_i))$ where the $x_i$ are independently drawn from $P_\mathcal{X}$ and $F \sim P_\mathcal{F}$, runs in polynomial time, and learns $(P_\mathcal{F}, P_\mathcal{X})$ with accuracy $\alpha$. Then there exists $\gamma > 0$, and a polynomial-sized neural net $(G_n, f)$ such that using perturbed stochastic gradient descent with noise $\delta$, learning rate $\gamma$, and loss function $L(x) = x^2$ to train $(G_n, f)$ on $t_n$ samples $((2x_i - 1, 2r_i - 1), 2F(x_i) - 1)$ where $(x_i, r_i) \sim P_\mathcal{X} \times B_{1/2}$ learns $(P_\mathcal{F}, P_\mathcal{X})$ with accuracy $\alpha - o(1)$.*

*Proof.* Let $A$ be an efficient algorithm that learns $(P_\mathcal{F}, P_\mathcal{X})$ with accuracy $\alpha$, and $t_n$ be a polynomial in $n$ such that $A$ uses fewer than $t_n$ samples and random bits with probability $1 - o(1)$. Next, define $h_n\{0,1\}^{(n+1)t_n + t_n + n \to \{0,1\}}$ such that the algorithm outputs $h_n(z_1, ..., z_{t_n}, b_1, ..., b_{t_n}, x')$ if it receives samples $z_1, ..., z_{t_n}$, random bits $b_1, ..., b_{t_n}$ and final input $x'$. There exists a polynomial $t_n^\star$ such that $A$ computes $h_n(z_1, ..., z_{t_n}, b_1, ..., b_{t_n}, x')$ in $t_n^\star$ or fewer steps with probability $1 - o(1)$ given samples $z_1, ..., z_{t_n}$ generated by a function drawn from $(P_\mathcal{F}, P_\mathcal{X})$, random bits $b_1, ..., b_{t_n}$, and $x' \sim P_\mathcal{X}$. So, let $h_n'(z_1, ..., z_{t_n}, b_1, ..., b_{t_n}, x')$ be $h_n(z_1, ..., z_{t_n}, b_1, ..., b_{t_n}, x')$ if $A$ computes it in $t_n^\star$ or fewer steps and $0$ otherwise. $h_n'$ can always be computed in polynomial time, so by the previous lemma there exists a polynomial sized neural net $(G_n, f)$ that gives an output within $1/2$ of $2h_n'((x_1, y_1), ..., (x_{t_n}, y_{t_n}), b_1, ..., b_{t_n}, x') - 1$ with probability $1 - o(1)$ when it is trained using noisy SGD with noise $\Delta$, learning rate $2^{716/3} 3^{24}$, and loss function $L$ on $((2x_i - 1, 2b_i - 1), 2F(x_i) - 1)$ and then run on $2x' - 1$. When the $(x_i, y_i)$ are generated by a function drawn from $(P_\mathcal{F}, P_\mathcal{X})$, and $x' \sim P_\mathcal{X}$, using $A$ to learn the function and then compute it on $x'$ yields $h_n'(z_1, ..., z_{t_n}, b_1, ..., b_{t_n}, x')$ with probability $1 - o(1)$. Therefore, training this net with noisy SGD in the manner described learns $(P_\mathcal{F}, P_\mathcal{X})$ with accuracy $\alpha - o(1)$. $\square$

**Remark 12.** *Like in the noise free case it would be possible to emulate a metaalgorithm that learns any function that can be learned from $n^c$ samples in $n^c$ time instead of an algorithm for a specific distribution. However, unlike in the noise free case there is no easy way to adapt the metaalgorithm to cases where we do not have an upper bound on the number of samples needed.*

**Remark 13.** *Throughout the learning process used by the last theorem and lemma, every control vertex, output control vertex, and vertex in the computation component always takes on a value where the activation function has derivative $0$. As such, the weights of any edges leading to these vertices stay within $\epsilon$ of their original values. Also, the conditions of lemma 4 are satisfied, so none of the edge weights in the memory component go above $\epsilon'$ more than double their original values. That leaves the edges from the output control vertices to the output vertex. Each output vertex only takes on a nonzero value once, and on that step it has a value of $2$. The derivative of the loss function with respect to the input to the output vertex is at most $12$, so each such edge weight changes by at most $24\gamma + \epsilon$ over the course of the algorithm. So, none of the edge weights go above a constant (i.e., $2^{242} 3^{25}$) during the training process.*

## 4.3 Additional comments on the emulation

The previous result uses choices of a neural net and SGD parameters that are in many ways unreasonable. This choice of activation function is not used in practice, many of the vertices do not have edges from the constant vertex, and the learning rate is deliberately chosen to be so high that it keeps overshooting the minima. If one wanted to do something normal with a neural net trained by SGD one is unlikely to do it that way, and using it to emulate an algorithm is much less efficient than just running the algorithm directly, so this is unlikely to come up.

In order to emulate a learning algorithm with a more reasonable neural net and choice of parameters, we will need to use the following ideas in addition to the ideas from the previous result. First of all, we can control which edges tend to have their weights change significantly by giving edges that we want to change a very low starting weight and then putting high weight edges after them to increase the derivative of the output with respect to them. Secondly, rather than viewing the algorithm we are trying to emulate as a fixed circuit, we will view it as a series of circuits that each compute a new

output and new memory values from the previous memory values and the current inputs. Thirdly, a lower learning rate and tighter restrictions on how quickly the network can change prevent us from setting memory values in one step. Instead, we initialize the memory values to a local maximum so that once we perturb them, even slightly, they will continue to move in that direction until they take on the final value. Fourth, in most steps the network will not try to learn anything, so that with high probability all memory values that were set in one step will have enough time to stabilize before the algorithm tries to adjust anything else. Finally, once we have gotten to the point that the algorithm is ready to approximate the function, its estimates will be connected to the output vertex, and the output will gradually become more influenced by it over time as a basic consequence of SGD.

## 5 Proofs of negative results

### 5.1 Proof of Theorem 3

Recall that for a sample set $S_m^{(t)} = \{X_1^{(t)}, \ldots, X_m^{(t)}\}$,

$$\hat{P}_{S_m^{(t)}} = \frac{1}{m} \sum_{i=1}^{m} \delta_{X_i^{(t)}} \tag{10}$$

and

$$W^{(t)} = W^{(t-1)} - \mathbb{E}_{X \sim \hat{P}_{S_m^{(t)}}} G_{t-1}(W^{(t-1)}(X), F(X)) + Z^{(t)}, \quad t = 1, \ldots, T. \tag{11}$$

*Proof of Theorem 3.* Consider running the descent algorithm on either true data labelled with $F$ or random data labelled with random labels, i.e.,

$$W_H^{(t)} = W_H^{(t-1)} - \mathbb{E}_{(X,Y) \sim D_{H,m}^{(t)}} G_{t-1}(W^{(t-1)}(X), Y) + Z^{(t)}, \quad t = 1, \ldots, T, \tag{12}$$

where

$$D_{H,m}^{(t)}(x, y) = \begin{cases} P_{S_m^{(t)}}(x)(1/2) & \text{if } H = \star, \\ P_{S_m^{(t)}}(x)\delta_{F(X)}(y) & \text{if } H = F. \end{cases} \tag{13}$$

Denote by $Q_H^{(t)}$ the probability distribution of $W_H^{(t)}$ and let $S_m^{(\leq t)} := (S_m^{(1)}, \ldots, S_m^{(t)})$. We then have the following.

$$\mathbb{P}\{W_F^{(T)}(X) = F(X)\} \leq \mathbb{P}\{W_\star^{(T)}(X) = F(X)\} + \mathbb{E}_{F, S_m^{(\leq T)}} d(Q_F^{(T)}, Q_\star^{(T)} | F, S_m^{(\leq T)})_{TV} \tag{14}$$

$$= 1/2 + \mathbb{E}_{F, S_m^{(\leq T)}} d(Q_F^{(T)}, Q_\star^{(T)} | F, S_m^{(\leq T)})_{TV}. \tag{15}$$

For $t \in [T+1]$ $H, h \in \{F, \star\}$, define

$$W_{H,h}^{(t-1)} = W_H^{(t-1)} - \left(\mathbb{E}_{(X,Y) \sim D_{h,m}^{(t)}} G_{t-1}(W_H^{(t-1)}(X), Y)\right) + Z^{(t)}, \tag{16}$$

and denote by $Q_{H,h}^{(t-1)}$ the distribution of $W_{H,h}^{(t-1)}$. Note that the above corresponds to taking one step using the data from $h$ after $t-1$ steps using the data from $H$.

Using the triangular and Data-Processing inequalities, we have

$$d(Q_F^{(t)}, Q_\star^{(t)} | F, S_m^{(\leq t)})_{TV} \tag{17}$$

$$\leq d(Q_{F,F}^{(t-1)}, Q_{\star,F}^{(t-1)} | F, S_m^{(\leq t)})_{TV} + d(Q_{\star,F}^{(t-1)}, Q_{\star,\star}^{(t-1)} | F, S_m^{(\leq t)})_{TV} \tag{18}$$

$$\leq d(Q_F^{(t-1)}, Q_\star^{(t-1)} | F, S_m^{(<t)})_{TV} + d(Q_{\star,F}^{(t-1)}, Q_{\star,\star}^{(t-1)} | F, S_m^{(\leq t)})_{TV} \tag{19}$$

$$= d(Q_F^{(t-1)}, Q_\star^{(t-1)} | F, S_m^{(<t)})_{TV} \tag{20}$$

$$+ TV(\mathbb{E}_{(X,Y) \sim D_{m,F}^{(t)}} G_{t-1}(W_\star^{(t-1)}(X), Y) + Z^{(t)}, \mathbb{E}_{(X,Y) \sim D_{m,\star}^{(t)}} G_{t-1}(W_\star^{(t-1)}(X), Y) + Z^{(t)} | F, S_m^{(\leq t)}). \tag{21}$$

Let $t$ be fixed, $S = (X, Y)$, $g(S) := G_{t-1}(W_\star^{(t-1)}(X), Y)$, $D. = D.^{(t)}$. We have by Pinsker's inequality[18],

$$TV(\mathbb{E}_{S \sim D_{m,F}} g(S) + Z^{(t)}, \mathbb{E}_{S \sim D_{m,\star}} g(S) + Z^{(t)} | F, S_m^{(\leq t)}, Z^{(<t)}) \leq \frac{1}{2\sigma} \|\mathbb{E}_{S \sim D_{m,F}} g(S) - \mathbb{E}_{S \sim D_{m,\star}} g(S)\|_2, \tag{22}$$

and by Cauchy-Schwarz,

$$\mathbb{E}_F TV(\mathbb{E}_{S \sim D_{m,F}} g(S) + Z^{(t)}, \mathbb{E}_{S \sim D_{m,\star}} g(S) + Z^{(t)} | F, S_m^{(\leq t)}, Z^{(<t)}) \tag{23}$$

$$\leq \frac{1}{2\sigma} (\mathbb{E}_F \|\mathbb{E}_{S \sim D_{m,F}} g(S) - \mathbb{E}_{S \sim D_{m,\star}} g(S)\|_2^2)^{1/2}. \tag{24}$$

We now investigate a single component $e \in E(G)$ in the above norm, i.e.,

$$\mathbb{E}_F (\mathbb{E}_{S \sim D_{m,F}} g_e(S) - \mathbb{E}_{S \sim D_{m,\star}} g_e(S))^2. \tag{25}$$

We have

$$\mathbb{E}_F (\mathbb{E}_{S \sim D_{m,F}} g_e(S) - \mathbb{E}_{S \sim D_{m,\star}} g_e(S))^2 = \mathbb{E}_F (\mathbb{E}_{S \sim D_{m,\star}} g_e(S)(1 - D_{m,F}(S)/D_{m,\star}(S)))^2 \tag{26}$$

$$= \mathbb{E}_F \langle g_e, (1 - D_{m,F}/D_{m,\star}) \rangle_{D_{m,\star}}^2 \tag{27}$$

$$= \mathbb{E}_F \langle g_e^{\otimes 2}, (1 - D_{m,F}/D_{m,\star})^{\otimes 2} \rangle_{D_{m,\star}^2} \tag{28}$$

$$= \langle g_e^{\otimes 2}, \mathbb{E}_F (1 - D_{m,F}/D_{m,\star})^{\otimes 2} \rangle_{D_{m,\star}^2} \tag{29}$$

$$\leq [(\mathbb{E}_{S \sim D_{m,\star}} g_e(S)^2) \|\mathbb{E}_F (1 - D_{m,F}/D_{m,\star})^{\otimes 2}\|_{D_{m,\star}^2}] \tag{30}$$

$$= (\mathbb{E}_{S \sim D_{m,\star}} g_e(S)^2)(\mathbb{E}_{S,S'} [\mathbb{E}_F (1 - D_{m,F}(S)/D_{m,\star}(S))(1 - D_{F,m}(S')/D_{m,\star}(S'))]^2)^{1/2} \tag{31}$$

$$= (\mathbb{E}_{S \sim D_{m,\star}} g_e(S)^2)(\mathbb{E}_{F,F'} [\mathbb{E}_{S \sim D_{m,\star}} (1 - D_{m,F}(S)/D_{m,\star}(S))(1 - D_{F',m}(S)/D_{m,\star}(S))]^2)^{1/2} \tag{32}$$

$$= (\mathbb{E}_{S \sim D_{m,\star}} g_e(S)^2) CP(m,t)^{1/2} \tag{33}$$

where (28) uses a tensor lifting to bring the expectation over $F$ on the second component before using the Cauchy-Schwarz inequality, and where (31) uses replicates, i.e., $(EZ)^2 = \mathbb{E}Z_1 Z_2$ for $Z, Z_1, Z_2$ i.i.d., with

$$CP(m,t) := \mathbb{E}_{F,F'} [\mathbb{E}_{S \sim D_{m,\star}^{(t)}} (1 - 2\delta_{F(X)}(Y))(1 - 2\delta_{F'(X)}(Y))]^2 \tag{34}$$

$$= \mathbb{E}_{F,F'} [\mathbb{E}_{X \sim P_{S_m^{(t)}}} F(X) F'(X)]^2. \tag{35}$$

Therefore, we have

$$\mathbb{E}_F TV(\mathbb{E}_{(X,Y) \sim D_{m,F}^{(t)}} G_{t-1}(W_\star^{(t-1)}(X), Y) + Z^{(t)}, \mathbb{E}_{(X,Y) \sim D_{m,\star}^{(t)}} G_{t-1}(W_\star^{(t-1)}(X), Y) + Z^{(t)} | F, S_m^{(\leq t)}, Z^{(<t)}) \tag{36}$$

$$\leq \frac{1}{\sigma} (\mathbb{E}_{S \sim D_{m,\star}^{(t)}} \|G(W_\star^{(t-1)}(X), Y)\|_2^2)^{1/2} CP(m,t)^{1/4} \tag{37}$$

and

$$TV(\mathbb{E}_{(X,Y) \sim D_{m,F}^{(t)}} G_{t-1}(W_\star^{(t-1)}(X), Y) + Z^{(t)}, \mathbb{E}_{(X,Y) \sim D_{m,\star}^{(t)}} G_{t-1}(W_\star^{(t-1)}(X), Y) + Z^{(t)}) \tag{38}$$

$$\leq \frac{1}{\sigma} \mathbb{E}_{S_m^{(\leq t)}, Z^{(<t)}} (\mathbb{E}_{S \sim D_{m,\star}^{(t)}} \|G_{t-1}(W_\star^{(t-1)}(X), Y)\|_2^2)^{1/2} CP(m,t)^{1/4}. \tag{39}$$

Defining the expected gradient norm as

$$GN(m,t) := \mathbb{E}_{S \sim D_{m,\star}^{(t)}} \|G_{t-1}(W_\star^{(t-1)}(X), Y)\|_2^2. \tag{40}$$

we get

$$\mathbb{E}_{F,S_m^{(\leq T)}} d(Q_F^{(T)}, Q_\star^{(T)} | F, S_m^{(\leq T)})_{TV} \leq \frac{1}{\sigma} \cdot \sum_{t=1}^{T} \mathbb{E}_{Z^{(<t)}} \mathbb{E}_{S_m^{(\leq t)}} (GN(m,t)^{1/2} \cdot CP(m,t)^{1/4}) \quad (41)$$

$$\leq \frac{1}{\sigma} \cdot \sum_{t=1}^{T} \mathbb{E}_{Z^{(<t)}} (\mathbb{E}_{S_m^{(\leq t)}} GN(m,t))^{1/2} \cdot (\mathbb{E}_{S_m^{(\leq t)}} CP(m,t)^{1/2})^{1/2}$$
$$(42)$$

$$= \frac{1}{\sigma} \cdot \sum_{t=1}^{T} \mathbb{E}_{Z^{(<t)}} (\mathbb{E}_{S_m^{(\leq t)}} GN(m,t))^{1/2} \cdot (\mathbb{E}_{S_m} CP(m,1)^{1/2})^{1/2}$$
$$(43)$$

and thus

$$\mathbb{E}_{F,S_m^{(\leq T)}} d(Q_F^{(T)}, Q_\star^{(T)} | F, S_m^{(\leq T)})_{TV} \leq \frac{1}{\sigma} CP_m^{1/4} \cdot \sum_{t=1}^{T} \mathbb{E}_{Z^{(<t)}} (\mathbb{E}_{S_m^{(\leq t)}} \mathbb{E}_{S \sim D_{m,\star}^{(t)}} \|G_{t-1}(W_\star^{(t-1)}(X), Y)\|_2^2)^{1/2}$$
$$(44)$$

$$= \frac{1}{\sigma} CP_m^{1/4} \cdot \sum_{t=1}^{T} \mathbb{E}_{Z^{(<t)}} (\mathbb{E}_{S_m^{(\leq t)}} \mathbb{E}_{S \sim D_\star^{(t)}} \|G_{t-1}(W_\star^{(t-1)}(X), Y)\|_2^2)^{1/2}$$
$$(45)$$

$$\leq \frac{1}{\sigma} CP_m^{1/4} \cdot \sum_{t=1}^{T} (\mathbb{E}_{Z^{(<t)}} \mathbb{E}_{S_m^{(\leq t)}} \mathbb{E}_{S \sim D_\star^{(t)}} \|G_{t-1}(W_\star^{(t-1)}(X), Y)\|_2^2)^{1/2}.$$
$$(46)$$

Finally note that

$$CP_m = \mathbb{E}_{F,F'} \mathbb{E}_{S_m} [\mathbb{E}_{X \sim P_{S_m}} F(X) F'(X)]^2 = 1/m + (1 - 1/m) CP_\infty. \quad (47)$$

$\square$

*Proof of Corollary 2.* GN is trivially bounded by $A^2 E$, so

$$\mathbb{E}_{F,S_m^{(\leq T)}} d(Q_F^{(T)}, Q_\star^{(T)} | F, S_m^{(\leq T)})_{TV} \leq \frac{A}{\sigma} E^{1/2} T(1/m + (1 - 1/m) CP_\infty)^{1/4}. \quad (48)$$

$\square$

### 5.1.1 Strengthening of Theorem 3 for parities

We now show the tighter bound with the term $CP^{1/2}$ rather than $CP^{1/4}$ for parities, which implies the following result that is a variant of the lower-bound from [Kea98] with slightly different exponents.

**Theorem 6.** *For each $n > 0$, let $(f, g)$ be a neural net of polynomial size in $n$. Run gradient descent on $(f, g)$ with less than $2^{n/10}$ time steps, a learning rate of at most $2^{n/10}$, Gaussian noise with variance at least $2^{-n/10}$ and overflow range of at most $2^{n/10}$. For all sufficiently large $n$, this algorithm fails at learning parities with accuracy $1/2 + 2^{-n/10}$.*

We first need the following inequalities.

**Lemma 7.** *Let $n > 0$ and $f : \mathbb{B}^{n+1} \to \mathbb{R}$. Also, let $X$ be a random element of $\mathbb{B}^n$ and $Y$ be a random element of $\mathbb{B}$ independent of $X$. Then*

$$\sum_{s \subseteq [n]} (\mathbb{E}f(X, Y) - \mathbb{E}f(X, p_s(X)))^2 \leq \mathbb{E}f^2(X, Y)$$

*Proof.* For each $x \in \mathbb{B}^n$, let $g(x) = f(x, 1) - f(x, 0)$.

$$\sum_{s \subseteq [n]} (\mathbb{E}[f(X,Y)] - \mathbb{E}[f(X, p_s(X))])^2 \tag{49}$$

$$= \sum_{s \subseteq [n]} \left( 2^{-n-1} \sum_{x \in \mathbb{B}^n} (f(x,0) + f(x,1) - 2f(x, p_s(x))) \right)^2 \tag{50}$$

$$= \sum_{s \subseteq [n]} \left( 2^{-n-1} \sum_{x \in \mathbb{B}^n} g(x)(-1)^{p_s(x)} \right)^2 \tag{51}$$

$$= 2^{-2n-2} \sum_{x_1, x_2 \in \mathbb{B}^n, s \subseteq [n]} g(x_1)(-1)^{p_s(x_1)} \cdot g(x_2)(-1)^{p_s(x_2)} \tag{52}$$

$$= 2^{-2n-2} \sum_{x_1, x_2 \in \mathbb{B}^n} g(x_1) g(x_2) \sum_{s \subseteq [n]} (-1)^{p_s(x_1)} (-1)^{p_s(x_2)} \tag{53}$$

$$= 2^{-2n-2} \sum_{x \in \mathbb{B}^n} 2^n g^2(x) \tag{54}$$

$$= 2^{-n-2} \sum_{x \in \mathbb{B}^n} [f(x,1) - f(x,0)]^2 \tag{55}$$

$$\leq 2^{-n-1} \sum_{x \in \mathbb{B}^n} f^2(x,1) + f^2(x,0) \tag{56}$$

$$= \mathbb{E}[f^2(X,Y)] \tag{57}$$

where we note that the equality from (51) to (54) is Parserval's identity for the Fourier-Walsh basis (here we used Boolean outputs for the parity functions). □

Note that by the triangular inequality the above implies

$$\mathrm{Var}_F \mathbb{E}_X f(X, F(X)) \leq 2^{-n} \mathbb{E}_{X,Y} f^2(X,Y). \tag{58}$$

As mentioned earlier, this is similar to Theorem 1 in [SSS17] which requires in addition the function to be the gradient of a 1-Lipschitz loss function.

We also mention the following corollary of Lemma 7 that results from Cauchy-Schwarz.

**Corollary 3.** *Let $n > 0$ and $f : \mathbb{B}^{n+1} \to \mathbb{R}$. Also, let $X$ be a random element of $\mathbb{B}^n$ and $Y$ be a random element of $\mathbb{B}$ independent of $X$. Then*

$$\sum_{s \subseteq [n]} |E[f((X,Y))] - E[f((X, p_s(X)))]| \leq 2^{n/2} \sqrt{E[f^2((X,Y))]}.$$

In other words, the expected value of any function on an input generated by a random parity function is approximately the same as the expected value of the function on a true random input.

*Proof of Theorem 6.* We follow the proof of Theorem 3 until (26), where we use instead Lemma 7, to write (for $m = \infty$)

$$\mathbb{E}_F (\mathbb{E}_{S \sim D_{m,F}} g_e(S) - \mathbb{E}_{S \sim D_{m,\star}} g_e(S))^2 \leq 2^{-n} \mathbb{E}_{S \sim D_{m,\star}} g_e^2(Z) \tag{59}$$

where $2^{-n}$ is the CP for parities. Thus in the case of parities, we can remove a factor of $1/2$ on the exponent of the CP. Further, the Cauchy-Schwartz inequality in (42) is no longer needed, and the junk-flow can be defined in terms of the sum of gradient norms, rather than taking norms squared and having a root on the sum; this does not however change the scaling of the junk-flow. The theorem follows by plugging the parameters of the statement. □

# 6 Some challenging functions

## 6.1 Parities

We start with the well-known problem of learning parities, which corresponds to $P_{\mathcal{X}}$ being uniform on $\{+1, -1\}^n$ and $P_{\mathcal{F}}$ being uniform on the set of parity functions defined by $\mathcal{P} = \{p_s : s \subseteq [n]\}$,

Figure 4: Two images of $13^2 = 169$ squares colored black with probability $1/2$. The left (right) image has an even (odd) number of black squares. The experiment illustrates the incapability of deep learning to learn the parity.

where $p_s : \{+1, -1\}^n \to \{+1, -1\}$ is such that

$$p_s(x) = \prod_{i \in s} x_i.$$

So nature picks $S$ uniformly at random in $2^{[n]}$, and with access to $\mathcal{P}$ but not to $S$, the problem is to learn which set $S$ was chosen from samples $(X, p_S(X))$ as defined in previous section.

Note that without noise or with low enough noise, this is *not* a hard problem. Even exact learning of the set $S$ (with high probability) can be achieved with an algorithm that builds a basis from enough samples (e.g., $n + \Omega(\log(n))$) and solves the resulting system of linear equations to reconstruct $S$.

This seems however far from how deep learning proceeds. For instance, descent algorithms are "memoryless" in that they update the weights of the NN at each step but do not a priori explicitly remember the previous steps. Since each sample (say for SGD) gives very little information about the true $S$, it thus seems unlikely for SGD to make any progress on a polynomial time horizon. However, it is not trivial to argue this formally if we allow the NN to be arbitrarily large and with arbitrary initialization (albeit of polynomial complexity), and in particular inspecting the gradient will typically not suffice. In fact, we showed that this is wrong, and SGD *can* learn the parity function with a proper initialization — See Section 4.

As discussed in Section 2, in the case of parities, our negative result for any initialization can be converted into a negative result for random initialization. We believe however that the randomness in a random initalization would actually be enough to account for any small randomness added subsequently in the algorithm steps. Namely, that one cannot learn parities with GD/SGD in poly-time with a random initialization.

To illustrate the phenomenon, we consider the following data set and numerical experiment in PyTorch [PGC$^+$17]. The elements in $\mathcal{X}$ are images with a white background and either an even or odd number of black dots, with the parity of the dots determining the label — see Figure 4. The dots are drawn by building a $k \times k$ grid with white background and activating each square with probability $1/2$.

We then train a neural network to learn the parity label of these images with a random initalization. The architecture is a 3 hidden linear layer perceptron with 128 units and ReLU non linearities trained using binary cross entropy. The training[19] and testing dataset are composed of 1000 images of grid-size $k = 13$. We used PyTorch implementation of SGD with step size 0.1 and i.i.d. rescaled uniform weight initialization [HZRS15].

Figure 5 show the evolution of the training loss, testing and training errors. As can be seen, the net can learn the training set but does not generalize better than random guessing.

## 6.2 Community detection and connectivity

Parities are not the most common type of functions used to generate real signals, but they are central to the construction of good codes (in particular the most important class of codes, i.e., linear codes, that rely heavily on parities). We mention now a few specific examples of

Figure 5: Training loss (left) and training/testing errors (right) for up to 80 SGD epochs.

functions that we believe would be also difficult to learn with deep learning. Connectivity is another notorious example discussed in the Perceptron book of Minsky-Papert [MP87]. In that vain, we provide here a different and concrete question related to connectivity and community detection. We then give another example of low cross-predictability distribution in arithmetic learning.

Consider the problem of determining whether or not some graphs are connected. This could be difficult because it is a global property of the graph, and there is not necessarily any function of a small number of edges that is correlated with it. Of course, that depends on how the graphs are generated. In order to make it difficult, we define the following probability distribution for random graphs.

**Definition 8.** *Given $n, m, r > 0$, let $AER(n, m, r)$ be the probability distribution of $n$-vertex graphs generated by the following procedure. First of all, independently add an edge between each pair of vertices with probability $m/n$ (i.e., start with an Erdős-Rényi random graph). Then, randomly select a cycle of length less than $r$ and delete one of its edges at random. Repeat this until there are no longer any cycles of length less than $r$.*

Now, we believe that deep learning with a random initialization will not be able to learn to distinguish a graph drawn from $AER(n, 10\ln(n), \sqrt{\ln(n)})$ from a pair of graphs drawn from $AER(n/2, 10\ln(n), \sqrt{\ln(n)})$, provided the vertices are randomly relabeled in the latter case. That is, deep learning will not distinguish between a patching of two such random graphs (on half of the vertices) versus a single such graph (on all vertices). Note that a simple depth-first search algorithm would learn the function in poly-time. More generally, we believe that deep learning would not solve community detection on such variants pruned-SBM models[20] (with edges allowed between the clusters as in a stochastic block model with similar loop pruning), as connectivity v.s. disconnectivity[21] is an extreme case of community detection.

The key issue is that no subgraph induced by fewer than $\sqrt{\ln(n)}$ vertices provides significant information on which of these cases apply. Generally, the function computed by a node in the net can be expressed as a linear combination of some expressions in small numbers of inputs and an expression that is independent of all small sets of inputs. The former cannot possibly be significantly correlated with the desired output, while the later will tend to be uncorrelated with any specified function with high probability. As such, we believe that the neural net would fail to have any nodes that were meaningfully correlated with the output, or any edges that would significantly alter its accuracy if their weights were changed. Thus, the net would have no clear way to improve.

## 6.3   Arithmetic learning

Consider trying to teach a neural net arithmetic. More precisely, consider trying to teach it the following function. The function takes as input a list of $n$ numbers that are written in base $n$ and are

$n$ digits long, combined with a number that is $n + 1$ digits long and has all but one digit replaced by question marks, where the remaining digit is not the first. Then, it returns whether or not the sum of the first $n$ numbers matches the remaining digit of the final number. So, it would essentially take expressions like the following, and check whether there is a way to replace the question marks with digits such that the expression is true.

$$120$$
$$+112$$
$$+121$$
$$=??0?$$

Here, we can define a class of functions by defining a separate function for every possible ordering of the digits. If we select inputs randomly and map the outputs to $\mathbb{R}$ in such a way that the average correct output is $0$, then this class will have a low cross predictability. Obviously, we could still initialize a neural net to encode the function with the correct ordering of digits. However, if the net is initialized in a way that does not encode the digit's meanings, then deep learning will have difficulties learning this function comparable to its problems learning parity. Note that one can sort out which digit is which by taking enough samples where the expression is correct and the last digit of the sum is left, using them to derive linear equation in the digits $\pmod{n}$, and solving for the digits.

We believe that if the input contained the entire alleged sum, then deep learning with a random initialization would also be unable to learn to determine whether or not the sum was correct. However, in order to train it, one would have to give it correct expressions far more often than would arise if it was given random inputs drawn from a probability distribution that was independent of the digits' meanings. As such, our notion of cross predictability does not apply in this case, and the techniques we use in this paper do not work for the version where the entire alleged sum is provided. The techniques instead apply to the above version, and GD-based deep learning cannot learn this arithmetic that is efficiently learnable by other means.

## Footnotes

[11]Two random bits will always be sufficient because the algorithm can spend as many timesteps as it needs copying random bits into memory and ignoring the rest of its input.

[12]This holds for any loss function that has a minimum when the output is correct, not just the $L_2$ loss function that we are using. We could avoid this by having the net output $\pm 1/2$ instead of $\pm 1$. However, if we did that then the change in each edge weight if the net got the right output would be $-1/3$ of the change in that edge weight if it got the wrong output, which would be likely to result in an edge weight that we did not want in at least one of those cases. There are ways to deal with that, but they do not seem clearly preferable to the current approach.

[13]Unless one uses more than two fan-in in the computation nodes, which can reduce the depth.

[14]Note that these will not be the input of the general neural net that is being built, but all the input entering the computation component besides from the constant vertex.

[15]It would be convenient if $v'_1, ..., v'_m$ all used the same encoding. However, the computation component will need to get inputs from the net's input vertices and from the memory component. The input vertices encode $0$ and $1$ as $\pm 1$, while the memory component encodes them as $\pm s'$ for some small $s'$. Therefore, it is necessary to be able to handle inputs that use different encodings.

[16]Note that these will not be the $n$ data input of the general neural net that is being built; these input vertices take both the data inputs and some inputs from the memory component.

[17]This time we can use the same values of $y^{(0)}$ and $y^{(1)}$ for all $v'_i$ because we just need them to be between whatever the vertex encodes 0 as and whatever it encodes 1 as for all vertices.

[18]One can get an additional $1/\pi$ factor by exploiting the Gaussian distribution more tightly.

[19]We pick samples from a pre-set training set v.s. sampling fresh samples; these are not expected to behave differently.

[20]It would be interesting to investigate the approach of [CLB17] on such models.

[21]Another similar example is to consider a single cycle v.s. two disjoint cycles on half of the vertices.