[Reviews · NeurIPS 2020]

Review 1

Summary and Contributions: The paper proves the following result. Given any polynomial time learning algorithm, that can learn some function class, it is possible to design some neural network architecture, such that gradient decent on this architecture will learn the aforementioned hypothesis class in polynomial time. This result show that neural network learning is universal, in the regime of efficient learning algorithms.

Strengths: This is really cool and basic observation, that is certainly worth publication

Weaknesses: The structure of the neural network is (not surprisingly) complex. Hence, it is not clear whether this result says something about neural networks that are used in practice.

Correctness: yes

Clarity: yes

Relation to Prior Work: yes

Reproducibility: Yes

Additional Feedback:


Review 2

Summary and Contributions: The paper shows that deep learning with SGD is a universal learning paradigm, i.e. for every problem P that is learnable using some algorithm A, there exists a poly size network and a corresponding initialization such that SGD learns P using poly step size in polynomial number of steps. Along this line, they show that SGD learns parities on a poly size network. Further, they provide a lower bound that shows that full gradient / large batch gradient descent is not a general learning paradigm. To the best of my knowledge this is the first work that shows this difference between GD and SGD at this level of generality.

Strengths: I think the paper aims to address a very fundamental problem in deep learning, and has useful implications to understand and compare various learning algorithms and NN architecture design. Please look at other answers for more details

Weaknesses: Order of the quantifiers: The main positive result essentially observes that if an algorithm A learns a problem P, then there exists a network and a corresponding initialization on which SGD will simulate the algorithm A on P. I am confused on the order of the quantifiers here. It seems that the network architecture depends on the algorithm A. Knowledge of the algorithm A will put some structure on the underlying learning problem P. Given that, it seems that the paper misuses the term “universal learning”. Ideally, one would want the network architecture and the initialization to be independent of the algorithm A or the problem P. Am I missing something here? Negative results for GD vs Optimization perspective: I am confused by the negative result. (a) The positive result suggests that there exists a network architecture for which there exists a set of weights that have a small generalization error. Suppose one starts with this network architecture, and does GD on the test loss. The negative results suggest that GD will fail to go to the global minima or a well generalizing stationary point. Can the authors please provide some intuition for why that would be the case for parities? From an optimization perspective, it seems that GD on the test loss with the right architecture should definitely succeed (at-least for a large set of values of initialization). (b) Typically the learning rate is chosen to be Poly(1/L, 1/T) where L is the gradient lipschitz parameter of the loss function. The lipschitz parameter may be exponentially large for poly sized networks, which would motivate a learning rate to be exponentially small (instead of polynomial or constant as is done in Corollary 1). Would GD with a small learning rate succeed? SQ lower bounds vs SGD: SQ algorithms can not learn parities (BKW03). It seems that SGD is an SQ algorithm? I am confused for why the lower bound for learning parities in the SQ model does not conflict with the positive results given in the paper.

Correctness: The proofs seem correct. I skimmed through it but did not verify everything line by line.

Clarity: The paper is mostly well written. However, I would recommend the authors to be more explicit about the order of the quantifiers, etc in the theorem statement. It might also help to break down some of the larger corollaries for ease of understanding.

Relation to Prior Work: Yes

Reproducibility: Yes

Additional Feedback: I would be flexible in changing my score depending on the feedback to my questions listed in the weakness section. ===== Post author feedback and discussions ======= The authors have addressed some of my questions. I am thus, increasing my score.


Review 3

Summary and Contributions: This paper show several results that deep learning by SGD or GD can learn any function class in polytime that can be learned in polytime by some algorithm. Specifically, it shows SGD with poly steps can robustly learn any function class under poly-noise on gradients. The results enhance the universality of deep NN.

Strengths: This paper study the universality of deep learning, especially the function classes defined on parities. It is a challenging problem and the results in this paper answers several meaningful questions in this field.

Weaknesses: Although the results in this paper have theoretical values, its practical values are weak. Can you do experiments to show that SGD can be more efficient than SQ on learning parities as claimed in item 3 in introduction?

Correctness: Yes

Clarity: Not very well. The theorems and proofs are hard to follow and some statements are not rigorous. For example, in the paragraph in front of theorem 2 in sec2.2, you say "inverse-poly noise" while in introduction, you say "polynomial noise can be added". Which one is accurate, inverse-poly or poly? Another example is "therefore the previous theorem is not a degeneracy due to infinite precision". What does "infinite precision" refer to? The introduction of SQ algorithm is necessary since there are many results compared with it. As there may be many existing results, it is better to provide a table to summary them and through the table, your contributions can be clearly clarified. It is better to provide a proof sketch with clear structure.

Relation to Prior Work: Yes

Reproducibility: Yes

Additional Feedback: 1. Can you explain more on the statement: "In a practical setting, there may be no reason to use SGD to a general learning algorithm"? As DL is universal, what is the drawback of SGD? 2. The results in Section 2.2 show that there exists an initialization for SGD to learn the target function. Are there any practical principle or algorithm to construct the initialization? 3. I am not familiar with stochastic query algorithm (SQ), so the claims about the relation between SQ and DL is not clear to me. It is better to introduce the setting of SQ in background. 4. Is it possible to characterize the exact order of "Poly" in the theoretical results? 5. One suggestion: cite the related theorem when you claim the contributions in introduction.

[Author Response · NeurIPS 2020]

**We thank all the reviewers for their useful comments and positive feedback.**

- *On the 'practicality':* it depends what 'practical' means. To be concrete, it would be **possible to implement the**
**positive scheme** (architecture and initialization) to learn parties on a laptop. This would require driving down the
constants as discussed in the appendix and implementing the scheme described in Example 1. It would not be a trivial
task -and not a major addition we believe- but it would be feasible (this is proposed as a senior thesis project).

Parities give an interesting case as one can't learn parities efficiently with SQ algorithms. Re R3: This goes thus
beyond improving on SQ; **one could simply not implement this with SQ.** Further one could not implement this either
with small enough neural nets (we need at least $n^2$ edges and currently depth $\log(n)$). So this stresses the power of
SGD-based deep learning. If allowed to be speculative, one may wonder whether NNs and SGD could eventually
become the components of general computational systems, in which case some of the mechanisms highlighted in
Section 2.4 may turn useful. Having said that, our goal was not to replace all other learning algorithms with DL
(our general emulation would loose efficiency compared to tailored algorithms in most cases), but rather to show that
**SGD-based deep learning has universality properties that are not granted to all learning paradigms**.

- *Further background on statistical query (SQ) algorithms:* Kearns introduced these to investigate which functions
could be learned when having access to 'statistics' about the function data. One particular example of statistics being
the gradient on the test loss (i.e., full GD). Kearns further introduced the key parameter of the precision noise (the
exact population distribution being not accessible). He then shows that not all functions are poly-time learnable with
poly-precision noise in the SQ framework, with parities as the prominent counter-example. This was after Minsky-Papert
noticed the issue of the perceptron with parities. Our contribution shows that **parities are actually learnable** with
**large enough** neural nets trained by **SGD**, with all relevant parameters being polynomial, including the precision noise
(i.e., without 'cheating'). In fact, **SGD is not an SQ algorithm** (R2), because the queries are **stochastic**, i.e., one picks
a random sample and not an average over the population distribution (SGD is what's sometimes called a 1-STAT oracle).

- *The vagueness around 'poly':* indeed we are in various place hand-wavy about 'poly' or 'inverse-poly'. We do not
view this as a lack of rigor because **there is always only one way that is meaningful**, i.e., if the noise were to be
polynomially large, it would be trivial to prove our negative results. However, we agree with the reviewers that this may
not necessarily be obvious to the reader, so we will re-write the statements to be explicit on the poly terms. E.g.: if the
batch-size is at least polynomially large ($n^c$ with $c$ large enough), the noise is at least inverse-polynomially large ($n^{-c'}$
with $c'$ small enough), $\mathrm{CP}_\infty$ is at least inverse-polynomially small ($n^{-c''}$ with $c''$ large enough) and the NF is at most
polynomially large ($n^{c'''}$ with $c'''$ small enough), then no matter what the net initialization and architecture are, learning
with accuracy $1/2 + \Omega_n(1)$ is not solvable. We will also put the exponents where we can, but some are tedious to get.

- *Regarding the precision:* it is important for this type of work to take into account the precision noise, i.e., one can
achieve degenerate result if able to work with infinite precision/magnitude on the edge weights, as for SQ algorithms.

- *On the universality notion (R2):* yes the quantifiers in Theorem 1 are in the order mentioned. One is of course not
given the function to be learned, but the class or distribution, and one can exploit this knowledge. Note that even in
this setting, **SQ or small enough nets would not achieve such a universality.** Further, as stated in Remark 3, this
implies as a direct corollary the result in the appendix that removes the knowledge of the class/algorithm and requires
only a bound on the polynomial complexity: one can build a net that matches the best asymptotic performance of any
algorithm that uses at most $n^c$ time and $n^c$ samples for any known $c$, **without knowing the algorithm in advance.** We
will move this result up in the main part with the extra page (if accepted). Further, it is necessary to know $c$ because
given a neural net of size $O(n^{c'})$ we could just pick a function that requires a net of size $\Omega(n^{c'+1})$ to compute.

- *Some 'insight' on the GD failure (R2):* Conceptually, when one trains a neural net using gradient descent one is
comparing the accuracy of the current net with the accuracies of the nets that would result from perturbing an edge
weight slightly, and then changing the weights in the direction of improved accuracy. This commonly works in practice
because most functions that one wants to learn are correlated with functions that the net is likely to compute. In such
cases full gradients are typically beneficial. However, **this intuition breaks down for certain functions like random**
**high-degree monomials:** functions that significantly correlate with random parities have vanishing probability. So, if
we try to learn such a random function using GD to train a neural net, it is likely that neither the original net nor any of
the nets resulting from shifting one edge's weight will be significantly correlated with the desired function, and the full
gradients will be almost independent of the function. Instead one needs to extract more details about the function on
individual samples in order to succeed with such functions, as the Gaussian elimination or SGD algorithms permit.

R2: No, GD with a small learning rate would not succeed in the considered setting. With an exponentially large learning
rate we could have the component of the gradient that actually corresponded to the parity we were trying to learn be
nonnegligable. We thus have to handle this technicality for the proof to hold, even though this is not a very relevant
scenario for applications. **For smaller learning rates, the failure is less difficult to obtain** and is part of our result.

[Meta-Review · NeurIPS 2020]

The paper shows that any functional class that can be learned in polynomial time by some algorithm can be learned in polynomial time by deep neural networks using stochastic gradient descent. This sheds light, in part, on the empirical success of deep learning, and makes an important contribution toward furthering our understanding of efficient learning of neural networks. Authors complement the result with several extensions, including (a) showing that the results hold even when polynomial noise is added to the gradients or when weights can be of polynomial precision, (b) showing that a network of size O(n^2) and depth O(log(n)) can learn parities using SGD in poly time, and (c) lower bounds for descent algorithms characterized in terms of novel properties of networks that may be of independent interest. The paper reads very well, and the results and insights in the paper are very compelling. Overall, a good paper. Accept!